# On the Price of Differential Privacy for Hierarchical Clustering

Chengyuan Deng [*]  Jie Gao [*]  Jalaj Upadhyay [*]  Chen Wang [†]  Samson Zhou [†]

## Abstract

Hierarchical clustering is a fundamental unsupervised machine learning task with the aim of organizing data into a hierarchy of clusters. Many applications of hierarchical clustering involve sensitive user information, therefore motivating recent studies on differentially private hierarchical clustering under the rigorous framework of Dasgupta's objective. However, it has been shown that any privacy-preserving algorithm under edge-level differential privacy necessarily suffers a large error. To capture practical applications of this problem, we focus on the weight privacy model, where each edge of the input graph is at least unit weight. We present a novel algorithm in the weight privacy model that shows significantly better approximation than known impossibility results in the edge-level DP setting. In particular, our algorithm achieves $O(\log^{1.5} n/\varepsilon)$ multiplicative error for $\varepsilon$-DP and runs in polynomial time, where $n$ is the size of the input graph, and the cost is never worse than the optimal additive error in existing work. We complement our algorithm by showing if the unit-weight constraint does not apply, the lower bound for weight-level DP hierarchical clustering is essentially the same as the edge-level DP, i.e. $\Omega(n^2/\varepsilon)$ additive error. As a result, we also obtain a new lower bound of $\tilde{\Omega}(1/\varepsilon)$[1] additive error for balanced sparsest cuts in the weight-level DP model, which may be of independent interest. Finally, we evaluate our algorithm on synthetic and real-world datasets. Our experimental results show that our algorithm performs well in terms of extra cost and has good scalability to large graphs.

## 1 Introduction

Hierarchical clustering is a fundamental problem in unsupervised machine learning that has gained significant popularity since its introduction in 1963 (Ward Jr, 1963). In contrast to flat clusterings, such as the popular $k$-means and $k$-median, which provide a single partitioning of the input dataset, hierarchical clustering (HC) recursively partitions the dataset so that similar items are grouped together at lower levels of granularity while dissimilar items are separated as high as possible. The resulting output is a binary tree-like structure called a *dendrogram*, whose leaves represent the individual items and whose internal nodes represent a cluster of the items in its subtree, thereby organizing the relationships within the dataset into a hierarchical representation. Moreover, unlike flat clusterings that require the "correct" number of clusters as input and which can be difficult to ascertain (Thorndike, 1953), hierarchical clustering simultaneously captures structure at all levels of granularity and thus does not need to determine a fixed number of clusters. Consequently, hierarchical clustering has numerous applications in areas where data intrinsically is organized into hierarchical structure, such as biology and phylogenetics (Sneath & Sokal, 1973; Sotiriou et al., 2003), community detection (Leskovec et al., 2020), finance (Tumminello et al., 2010), and image and text analysis (Steinbach et al., 2000).

Even though these applications involve datasets with sensitive information, there has been little work on privacy-preserving hierarchical clustering. Imola et al. (2023) recently initiated a study on differentially-private approximation algorithms for hierarchical clustering measured using the objective introduced by Dasgupta (2016), which quantifies the HC cost based on how early similar nodes are split in the hierarchy. In particular, the cost is the sum of the weights of all edges, multiplied

---

[*]Rutgers University, {`cd751`,`jg1555`,`jalaj.upadhyay`}@rutgers.edu

[†]Texas A&M University, {`jhwjhw0123`, `samsonzhou`}@gmail.com

[1]Here, and throughout, we use $\tilde{O}()$ and $\tilde{\Omega}()$ to hide polylog$n$ terms.

by the size of the smallest cluster that contains both endpoints of the edge. Imola et al. (2023) considered *edge-level differential privacy*, where the edges of the input graph represent the sensitive information so that the output of the hierarchical clustering algorithm should be somewhat insensitive to changes to a single edge of the similarity graph. Intuitively, this is quite difficult to guarantee, since many common hierarchical clustering algorithms, such as *single-linkage* or *average-linkage* (Hastie et al., 2009) are deterministic and thus can change drastically when a single edge of the graph is changed. Indeed, Imola et al. (2023) showed that any $\varepsilon$-differentially private algorithm on a graph with $n$ vertices would produce a hierarchical clustering with Dasgupta's objective value $\Omega\left(n^2/\varepsilon\right)$ additively worse than the optimal clustering.

Such a large error renders the corresponding hierarchical clustering useless in most practical scenarios. Consider, for example, a dataset whose nonzero costs construct a graph that consists of $\sqrt{n}$ disconnected instances of an expander graph with $(\log n)$-expansion (Kowalski, 2019) and on $\sqrt{n}$ vertices. We remark that each expander can incur a cost at most $(\sqrt{n})^2 \log n$ under the Dasgupta objective, and so by clustering each of the $\sqrt{n}$ expanders together, the optimal hierarchical clustering cost is at most $n^{1.5} \log n$. On the other hand, if we have a single connected $(\log n)$-expander on $n$ vertices, then it can be shown that any balanced cut at the top level of the dendrogram incurs must have $O(\log n)$ edges that cross the cut. Since it is known that there exists a balanced cut that is a constant-factor approximation to the optimal hierarchical clustering (Charikar & Chatziafratis, 2017; Roy & Pokutta, 2017), it follows that the optimal cost in this case is $\Omega(n^2 \log n)$. Hence, algorithms with additive error $\Omega\left(n^2/\varepsilon\right)$ will not be able to distinguish between these two cases.

In light of these shortcomings, it is natural to ask whether more meaningful accuracy-privacy tradeoffs are possible under less stringent notions of differential privacy. Another common notion is *weight-level* differential privacy, which assumes that the graph topology and connectivity are public knowledge but edge weights are sensitive. Therefore, the private mechanism in this scenario bounds the probability of distinguishing two neighboring graphs with the same connectivity, but different weights, where a pair of neighboring graphs have weight vectors that differ by an $\ell_1$ norm of 1. Weight-level differential privacy is a great fit for many application scenarios where the underlying network topology is often public, e.g., Internet topology or financial networks. However, the weights on such networks may represent information that is closely tied to user behaviors and activities and should be protected. Hence, weight-level differential privacy has been studied for a number of fundamental problems such as all-pairs shortest paths and range queries Bodwin et al. (2024); Chen et al. (2023); Deng et al. (2023); Fan et al. (2022); Fan & Li (2022); Sealfon (2016).

**Our Results.** In this paper, we provide a comprehensive treatment for hierarchical clustering under Dasgupta's objective function with weight-level DP. To overcome the strong barrier in the edge-DP model, we consider the weight-DP model with a simple and rather unassuming setting where all edges have minimum weight 1. We show there exists an algorithm in this setting that achieves $O(\log^{1.5} n/\varepsilon)$ multiplicative approximation. As a complement of the upper bound, we next show that, surprisingly, the $\Omega(n^2/\varepsilon)$ lower bound still holds in the weight-DP model without the assumption. Finally, as a by-product of our HC results, we provide new lower bounds for weight-DP balanced sparsest cuts, which is a related problem where the objective is to partition a graph into two sets with a minimal "relative" number of crossing edges. Our results highlight the power and limitations of the weight-level DP model for HC and related problems. We now describe these results in more detail, starting with our new algorithm.

> **Result 1** (Informal Statement of Theorem 1). *There exists an $\varepsilon$-weight DP algorithm that given a weighted, connected graph $G$ of size $n$ with weight on each edge at least 1, in polynomial time computes an HC $\mathcal{T}$ that achieves an $O(\frac{\log^{1.5} n}{\varepsilon})$-multiplicative approximation for the optimal HC cost.*

Our private algorithm gives roughly $O(\log^{1.5} n)$-approximation to the optimal cost. In the HC literature, $\mathrm{polylog}(n)$-approximation is considered to be a "sweet spot" solution (see, e.g. Dasgupta (2016); Charikar & Chatziafratis (2017); Roy & Pokutta (2017); Agarwal et al. (2022); Assadi et al. (2022)). In practice, it allows us to "adjust" to specific instances, and produce HC trees tailored to the actual optimal cost. Compared to the poly-time algorithm in Imola et al. (2023), which has $O\left(\frac{n^{2.5}\sqrt{\log(1/\delta)}}{\varepsilon}\right)$ additive error for $(\varepsilon, \delta)$-DP, our algorithm achieves $\varepsilon$-DP and has much better performance on the aforementioned one-vs-many expander example.

We will discuss shortly why the natural assumption for weight-level DP is the "right notion" for many privacy applications to obtain low approximation error on HC. Here, we first show that from a technical point of view, the unit-weight assumption is *necessary*.

> **Result 2** (Informal Statement of Theorem 3). *Any $\varepsilon$-weight DP algorithm that outputs an HC tree under Dasgupta's cost must have an additive error of $\Omega(n^2/\varepsilon)$, even for graphs with optimal HC cost $O(n/\varepsilon)$.*

Our lower bound for weight-level DP matches the lower bound of $\Omega(n^2/\varepsilon)$ for edge-level DP (Imola et al., 2023). Thus, our result precludes any practical algorithms for general graphs for either weight or edge-level DP. We present the formal theorem and the proof in Appendix C. As a result of the lower bound in Theorem 3, we also obtain a new lower bound for balanced sparsest cut for weight-level DP.

> **Result 3** (Informal Statement of Theorem 2). *Given a graph $G$ and a constant $\gamma$, any $\varepsilon$-weight DP algorithm that outputs a $\gamma$-balanced sparsest cut has to have $\Omega(1/(\varepsilon \log^2 n))$ expected additive error to the sparsity of $G$.*

We emphasize that Theorem 2 does *not* follow trivially from known connections between balanced sparsest cut and hierarchical clustering. The reason is: **(i)** known results deal with only multiplicative approximation, and **(ii)** we need to worry about the privacy loss with repeated balanced sparsest cuts. As such, we used a novel reduction to prove Theorem 2, which may be of independent interest.

**Discussions for our privacy model.** We discuss why our privacy model is natural and the contexts of its applications. For applications whose edge weights are not at least 1, we can scale up all weights by the same factor, and the combinatorial structure of the solutions to the HC does not change (although the cost of HC is also scaled by the same factor). Therefore, one can assume by default that the minimum edge weight is 1 when neighboring graphs have $\ell_1$ norm of edge weights at most 1. This assumption also makes sense as the definition of neighboring graphs in the differential privacy model typically assumes that the magnitude of change is at an "atomic" level.

Furthermore, the model captures a wide range of applications. For instance, consider the financial network where the edge weights capture the transaction interactions or amounts, and users would like to protect the transaction information. This scenario is captured exactly by our weight-DP model with unit weight assumption: (i) the topology is public to the network hosts, and (ii) the weight is at least one since each interaction contributes one and a reasonable transaction amount would be larger than one. Another application is clustering in music libraries, e.g., the iTunes library. The iTunes library is a bipartite graph where music/artists are on one side and users are on the other side. We know the graph topology from the album brought by the users. However, the number of times a user has interacted with a particular song is private information. Here, the interaction counter starts with 1 because a user has downloaded the album.

Finally, we note that our algorithm in Result 1 is *robust* for graphs that do *not* satisfy the unit-weight condition. For graphs with minimum weights $\alpha \leq 1$, our algorithm could achieve $O(\frac{\log^{1.5} n}{\alpha \cdot \varepsilon})$-multiplicative approximation with $\varepsilon$-weight DP. The discussion and formal analysis of this approximation guarantee is shown in Proposition E.1.

**Empirical evaluations.** In correspondence to Result 1, we implement the weight-DP algorithm for hierarchical clustering and evaluate its performance with respect to Dasgupta's cost. We observe that our algorithm produces significantly lower cost than input perturbation consistently on synthetic and real-world graphs. We further remark that our algorithm in Result 1 is the first DP hierarchical clustering algorithm with *reasonable implementations* on general graphs. Previously, the algorithms in Imola et al. (2023) for general graphs either take exponential time (the $\varepsilon$-DP algorithm) or use very complicated theoretical subroutines for which no implementation is known (the $(\varepsilon, \delta)$-DP algorithm). In contrast, our algorithm is practical based on favorable performance and good scalability. Our codes are publicly available on the Anonymous Github[2].

## 1.1 OUR TECHNIQUES

**The algorithm with unit weights.** The most natural approach to consider for our privacy setting is input perturbation. Naively, we would try to add Laplacian noise of $\mathsf{Lap}(\frac{1}{\varepsilon})$ to every edge weight to

---

[2]https://anonymous.4open.science/r/dp-hc-8612/

obtain graph $\widetilde{G}$. Then, we can perform recursive balanced sparsest cuts on the $\widetilde{G}$ to obtain an HC tree $\mathcal{T}$. The hope here is that on the balanced sparsest cut $S^*$, since each edge has at least unit weights, the noise scales at most with the sparsity.

Unfortunately, the simple idea as above does *not* work: although the balanced sparsest cut on $G$ remains a balanced sparse cut on $\widetilde{G}$, the *converse* is not true. In particular, let $(S, V \setminus S)$ be a cut with high sparsity in $G$, and suppose $E(S, V \setminus S)$ are of unit weights; then, because of the Laplacian noise, the sparsity of $S$ in $\tilde{G}$ can be very small. The same issue also happens if we instead list all the possible cuts, add $\mathsf{Lap}(\frac{\phi_G}{\varepsilon})$ noise to each entry ($\phi_G$ is the global sparsity of $G$), and output the sparsest cut therein. Here, since we have $2^n$ many balanced cuts, some of the cuts could have very large deviations in the noisy vector, which again breaks the utility.

Our idea to handle the above issue is to take advantage of the unit-weight condition, and "amplify" the gap between sparse and non-sparse cuts in $G$. To this end, we can use a fairly simple trick: we artificially "bump up" the weight of each edge in $G$ by $O(\frac{\log n}{\varepsilon})$ before adding the Laplacian noise. In this way, cuts with high sparsity will never get a chance to become low from the Laplacian noise. Furthermore, because of the unit-weight assumption, the actual sparsest cut $S^*$ will *not* be affected by too much due to the increased edge weights (at most an $O(\frac{\log n}{\varepsilon})$ factor). As such, we can perform recursive balanced sparsest cuts and get the desired algorithm.

**The lower bound for HC in weight-level DP.** Our lower bound is inspired by the $\varepsilon$-DP hierarchical clustering lower bound in Imola et al. (2023) for edge-level DP; however, generalizing the lower bound to our privacy model requires non-trivial additional work. The hard instances in Imola et al. (2023) come from the family of random disjoint 5-cycles. It is well-known in the literature that to minimize Dasgupta's HC cost on disconnected graphs, an HC tree should *avoid* splitting any edges before the induced subgraph becomes connected. In particular, for the family of random 5-cycles, if the HC tree splits the cycle edges in the bottom layers of the tree, the induced cost is at most $O(n)$. On the other hand, if the HC tree starts with partitioning $\Omega(n)$ edges in the cycles, the induced cost is at least $\Omega(n^2)$. The key lemma of Imola et al. (2023) shows that any algorithm that is $\varepsilon$-DP has to, unfortunately, cut many of the cycle edges, which leads to the desired lower bound of $\Omega(n^2)$.

For the weight-level privacy setting, we need to argue that the topology information revealed to the algorithm does not break the lower bound. This is not true at first glance: for any *fixed* collection of 5 cycles, if the algorithm knows the graph topology, it can simply avoid all the cycle edges while being $\varepsilon$-DP. Our idea to overcome the challenge is to "embed" the family of random 5 cycles into a complete graph. In particular, we can fix the graph topology as the complete graph and put "important weights" only on edges obtained by random 5 cycles. In this way, we essentially preserve the source of hardness: the algorithm cannot be private if it always avoids the edges with high weights.

Due to space limits, we defer the technical overview of the balanced sparsest cut lower bound to Appendix F.

## 2 PRELIMINARIES

We use $\mathbb{R}$ to denote the set of real numbers and $\mathbb{R}_{\geq 0}$ denote the set of non-negative real numbers. We use the notation $G = (V, E, w)$ to denote a graph on the vertex set $V$, edge set $E$, and weight function $w : V \times V \to \mathbb{R}_{\geq 0}$. The edge between two vertices $u$ and $v$ is denoted by the tuple $(u, v)$ and its weight is denoted by $w(u, v)$. For a set of vertices $A, B \subseteq V$, we denote by

$$w_G(A, B) = \sum_{(u,v) \in A \times B} w(u, v).$$

When the context is clear, we simplify it by writing $w(A, B)$. We now introduce Dasgupta's framework for hierarchical clustering.

**Definition 1** (Hierarchical clustering trees). Given a weighted graph $G = (V, E, w)$ with $n$ vertices representing data points and $m$ edges with non-negative weights measuring the pairwise similarities, we say a rooted tree $\mathcal{T}$ is a hierarchical clustering tree (HC tree) if the leaf nodes correspond to vertices $V$, and the internal nodes represent the splits of the subsets of vertices.

**Definition 2** (Hierarchical Clustering under Dasgupta's Objective (Dasgupta, 2016)). Given a weighted graph $G = (V, E, w)$, Dasgupta's HC objective is to minimize the cost of $\mathcal{T}$ prescribed as

follows:

$$\mathsf{cost}_G(\mathcal{T}) = \sum_{(u,v) \in E} w(u,v) \cdot |\mathcal{T}[u \vee v]|, \tag{1}$$

where $\mathcal{T}[u \vee v]$ stands for the subtree rooted at the lowest common ancestor of $u, v$, and $|\mathcal{T}[u \vee v]|$ is the number of *leaf nodes* induced by $\mathcal{T}[u \vee v]$. The clustering is represented by $\mathcal{T}$, where each internal node induces a cluster.

Prior works (Charikar & Chatziafratis, 2017; Cohen-Addad et al., 2019; Dasgupta, 2016) have established a connection between the (balanced) sparsest cut problem (Definition 9) and hierarchical clustering problem. In short, as long as the graph has non-negative weights, an $\alpha$-approximation of the sparsest cut or balanced sparsest cut implies an $O(\alpha)$-approximation algorithm for the hierarchical clustering. The formal statement of such an algorithm is given in Proposition A.6.

A variety of privacy models have been studied for graph-theoretic problems with differential privacy. The key difference lands on the definition of *neighboring graphs*, which determines the sensitive information to be protected by DP. Towards this end, most problems are studied under one or several of the following three privacy models: node-level (Blocki et al., 2013; Kalemaj et al., 2023; Kasiviswanathan et al., 2013; Sealfon & Ullman, 2021), edge-level (Arora & Upadhyay, 2019; Blocki et al., 2012; Dalirrooyfard et al., 2024; Eliáš et al., 2020; Gupta et al., 2012; Imola et al., 2023; Liu et al., 2024) and weight-level (Chen et al., 2023; Deng et al., 2023; Fan et al., 2022; Fan & Li, 2022; Sealfon, 2016). The focus of this study is the weight-level DP.

**Definition 3** (Neighboring weights). For a graph $G = (V, E)$, let $w, w' : V \times V \to \mathbb{R}_{\geq 0}$ be two weight functions that map any $(u, v) \in E$ to a non-negative real number, we say $w, w'$ are neighboring, denoted as $w \sim w'$ if: $\sum_{(u,v) \in E} |w(u,v) - w'(u,v)| \leq 1$.

Now we can formally define weight-differential privacy on a graph $G$.

**Definition 4** (Differential Privacy). An algorithm $\mathcal{A}$ is $(\varepsilon, \delta)$-DP on a graph $G = (V, E)$, if for any neighboring weights $w \sim w'$ such that $G' = (V, E, w')$, and any set of output $\mathcal{C}$, it holds that:

$$\Pr[\mathcal{A}(G) \in \mathcal{C}] \leq e^\varepsilon \cdot \Pr[\mathcal{A}(G') \in \mathcal{C}] + \delta.$$

If $\delta = 0$, we say $\mathcal{A}$ is $\varepsilon$-differentially private on $G$.

More standard technical tools in differential privacy are deferred to Appendix A.

## 3 PRIVATE ALGORITHM FOR HIERARCHICAL CLUSTERING

We start by formalizing our privacy model rooted in the weight-level DP that we believe is more appropriate for hierarchical clustering. This model has two assumptions in addition to neighboring weights, formally as below.

**Definition 5** (Neighboring weights). For a graph $G = (V, E)$ and weight functions $w, w' : E \to \mathbb{R}^{\geq 0}$, we say $w, w'$ are neighboring, denoted as $w \sim w'$ if: (1) $G$ is a connected component. (2) For any $e \in E$, $w(e) \geq 1$. (3) $\sum_{e \in E} \|w(e) - w'(e)\| \leq 1$.

We remark that the model in Definition 5 is a natural extension of the weight-neighboring notion in Definition 3, as discussed in Section 1. Our main algorithm for HC under this model is as follows.

**Theorem 1** (Formalization of Result 1). *There exists a polynomial-time $\varepsilon$-weight DP algorithm for any $\varepsilon > 0$, such that given a weighted, connected graph $G$ of size $n$ with weight on each edge at least 1, outputs a balanced HC tree $\mathcal{T}$ such that the HC cost by $\mathcal{T}$ is at most $O(\frac{\log^{1.5} n}{\varepsilon}) \cdot \mathsf{OPT}_G$, where $\mathsf{OPT}_G$ is the optimal HC cost of $G$ under Dasgupta's objective.*

A few remarks are in order. If exponential time is allowed, we can achieve $O(\frac{\log n}{\varepsilon})$ multiplicative approximation *or* $O(\frac{n^2 \log n}{\varepsilon})$ additive error. Furthermore, by using an algorithm that could privately evaluate the HC tree costs, we could augment our algorithm with the algorithm in Imola et al. (2023) to obtain an $(\varepsilon, \delta)$-DP algorithm with a cost at most

$$\min\left\{O(\frac{\log^{1.5} n \sqrt{\log(1/\delta)}}{\varepsilon}) \cdot \mathsf{OPT}_G + O(\frac{n \log n \sqrt{\log(1/\delta)}}{\varepsilon}), O(\sqrt{\log n}) \cdot \mathsf{OPT}_G + O(\frac{n^{2.5} \log^2 n \log^2(1/\delta)}{\varepsilon})\right\},$$

which strictly improves the bound of Imola et al. (2023). Finally, if the graph has minimum weight of $\alpha \in (0, 1]$, our algorithm could still achieve $\varepsilon$-weight privacy with $O(\frac{\log^{1.5} n}{\varepsilon \cdot \alpha})$-approximation error (see Proposition E.1 for details). We refer keen readers to Appendix E for such discussions; the rest of this section is dedicated to proving Theorem 1.

## 3.1 A Polynomial-time $\varepsilon$-DP Algorithm for Balanced Sparsest Cut

We prove Theorem 1 by showing a polynomial-time $\varepsilon$-DP algorithm for the balanced sparsest cut problem with multiplicative error at most $O(\log^{1.5}(n)/\varepsilon)$, under the weight-level privacy model following Definition 5. Our algorithm follows a simple procedure. First, overlay the input graph $G$ by adding an extra additive weight of $O(\log n/\varepsilon)$. Second, add independent Laplace noise $\mathsf{Lap}(1/\varepsilon)$ to every edge weight. Third, run a balanced sparsest cut algorithm (in the classical setting) on the perturbed graph and output the cut. Privacy of the mechanism is guaranteed by the Laplace mechanism used in the second step, also known as *input perturbation*, and the first step ensures that, with high probability, all the edge weights are positive. Therefore, we are allowed to run any sparsest cut algorithm by the post-processing theorem (Proposition A.2) in the third step. The output cut has a relatively small error because the magnitude of the perturbation is also small.

We still have to specify an algorithm to produce the balanced sparsest cut. It is known that a relaxation of this problem admits an $O(\sqrt{\log n})$-approximation when all edge weights are non-negative.

**Proposition 3.1** (Charikar & Chatziafratis (2017); Krauthgamer et al. (2009), rephrased)**.** *There exists an algorithm that given a graph $G = (V, E, w)$ such that $w(e) \geq 0$ for all $e \in E$, returns an $O(\sqrt{\log n})$ approximation to the balanced sparsest cut problem in polynomial time. Here, $n$ is the size of the input graph.*

Our algorithm is given as BALANCEDSPARSESTCUT-DP and uses polynomial time. We remark that it is possible to adapt our algorithm to further improve the utility bound by a multiplicative $O(\sqrt{\log n})$ factor, by using an exponential-time algorithm that returns the optimal value of the balanced sparsest cut, rather than the polynomial-time algorithm in the third step of algorithm. However, the resulting algorithm would then also use exponential time.

---

BALANCEDSPARSESTCUT-DP**: An $\varepsilon$-DP Algorithm for Balanced Sparsest Cut**
**Input:** Graph $G = (V, E, w)$, $|V| = n$, privacy parameter $\varepsilon > 0$.

1. $\forall e \in E$, add $10 \log n/\varepsilon$ to the edge weight $w(e)$, denoted as $G' = (V, E, w')$

2. $\forall e \in E$, add independent noise $\sim \mathsf{Lap}(1/\varepsilon)$ to $w'(e)$, denoted as $G'' = (V, E, w'')$.

3. Run the algorithm from Proposition 3.1 on $G''$ and return the cut $S$.

---

Now we provide the analysis of BALANCEDSPARSESTCUT-DP. Due to space limits, we defer some proofs to Appendix D.

**Lemma 3.1.** *The* BALANCEDSPARSESTCUT-DP *algorithm is $\varepsilon$-DP.*

Next, we show the proof of the multiplicative utility bound, captured by the lemma below.

**Lemma 3.2.** *With high probability, for any graph $G = (V, E, w)$ whose minimum weight is at least $1$, the* BALANCEDSPARSESTCUT-DP *algorithm returns a balanced cut in polynomial-time with at most $O(\log^{1.5} n/\varepsilon)$-approximation to the sparsity of the balanced sparsest cut.*

## 3.2 The Complete $\varepsilon$-DP Algorithm for Hierarchical Clustering

We present the $\varepsilon$-DP hierarchical clustering algorithm using the $\varepsilon$-DP balanced sparsest cut algorithm we discussed in the above section. As well-known in the literature, the sparsest cut algorithm is recursively called to construct an HC tree. The process, first proved by Dasgupta Dasgupta (2016), can be shown as follows.

---

MAKETREE: **Recursive Sparsest Cut for Hierarchical Clustering**
**Input:** Graph $G = (V, E, w)$
- If $|V| = 1$, return leaf containing $V$. Otherwise,
    - Run the algorithm in Proposition 3.1 with input $(G)$, let the output be $S$.
    - Let $G_S$ be the induced subgraph by $S$, similarly $G_{\bar{S}}$ for $\bar{S}$.
    - Run MAKETREE$(G_S)$ and MAKETREE$(G_{\bar{S}})$.

---

Now we are ready for the complete private algorithm for hierarchical clustering.

---

HIERARCHICALCLUSTERING-DP: **An $\varepsilon$-DP Algorithm for Hierarchical Clustering**
**Input:** Graph $G = (V, E, w)$, privacy parameter $\varepsilon > 0$.
- Run Steps 1 and 2 of BALANCEDSPARSESTCUT-DP with input $G$ and $\varepsilon$, let the perturbed graph be $G''$.
- Run MAKETREE$(G'')$.

---

**Privacy guarantee of** HIERARCHICALCLUSTERING-DP. For the purpose of private hierarchical clustering, since *all* cuts in the vertex-induced subgraphs of $G''$ is a function of $G''$, we can apply the post-processing theorem (Proposition A.2) to show that the our algorithm remains $\varepsilon$-DP. Note that this is how we avoid composition and any potential error blow-up therein.

**Multiplicative approximation guarantee of** HIERARCHICALCLUSTERING-DP. We have shown by Claim D.1 that $G''$ has all edges with non-negative weights with high probability, therefore the MAKETREE involving the balanced sparsest cut algorithm can proceed without violations. For the multiplicative approximation on *every* partition, we can perform the analysis of Lemma 3.2 on each vertex-induced subgraph. There are at most $O(n)$ internal nodes, so we can apply a union bound to make Lemma 3.2 hold with high probability throughout the process of HIERARCHICALCLUSTERING-DP. This ensures we can run $O(\log^{1.5} n/\varepsilon)$ approximation on all internal nodes, and obtain the desired multiplicative approximation by Proposition A.6.

## 4 EXPERIMENTAL EVALUATIONS

We implement our new algorithm HIERARCHICALCLUSTERING-DP and demonstrate the experimental results in this section. First, we evaluate the performance of our algorithm in terms of Dasgupta's objective on synthetic and real-world graphs. Next, we show our algorithm has favorable scalability with large graphs ($n \geq 1000$).

**Datasets and baselines.** Following a sequence of previous works (Cohen-Addad et al., 2017; Abboud et al., 2019; Manghiuc & Sun, 2021; Laenen et al., 2023), we generate two datasets of synthetic graphs from the stochastic block model (SBM) and hierarchical SBM (HSBM), then we select real-world datasets: IRIS, WINE and BOSTON from the scikit-learn package (Pedregosa et al., 2011). Some basic statistics of the datasets used can be found in Table 2 in Appendix H. For baseline methods satisfying weight-DP, we first consider *input perturbation*, which adds $\mathsf{Lap}(1/\varepsilon)$ random noise to each edge weight then applies the recursive sparsest cut algorithm on the perturbed graph. Input perturbation is a simple mechanism achieving DP but could possibly lead to $O(n^3)$ additive error to the Dasgupta's cost in the worst case. Next, we include single, average and complete linkage methods, which are widely used in practice for hierarchical clustering. We apply the same perturbation scheme as our proposed algorithm for a fair comparison. Finally, we include the non-private cost for completeness.

### 4.1 PERFORMANCE ON DASGUPTA'S COST

**Synthetic graphs.** For graphs from both SBM and HSBM, we give edge weights by sampling uniformly at random between 1 and 10. The results shown in this section are based on 10 different

graphs generated by the same set of parameters where $p = 0.7$ (intra-probability) and $q = 0.1$ (inter-probability). We provide results on a wider range of parameters for SBM and HSBM in Appendix H. As for the choice of $\varepsilon$, the privacy parameter, we test all algorithms on $\varepsilon \in \{0.01, 0.1, 0.5, 1, 2\}$. Note that $\varepsilon > 1$ is already considered as a weak privacy guarantee.

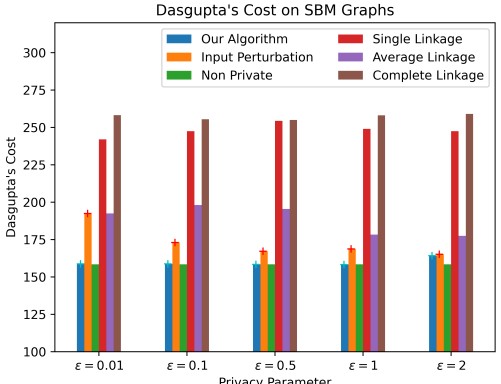 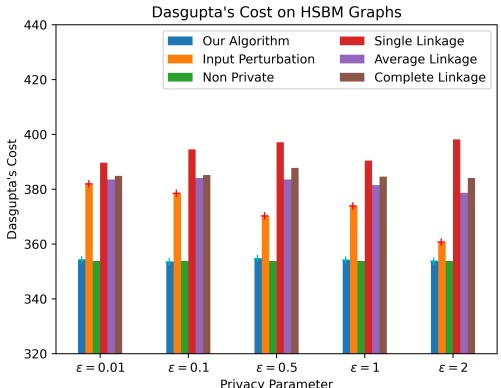

Figure 1: Comparison of Dasgupta's cost on SBM graphs of size $n = 150$ and $k = 5$.

Figure 2: Comparison of Dasgupta's cost on HSBM graphs of size $n = 150$ and $k = 5$.

Our results are shown in Figure 1 for SBM graphs and Figure 2 for HSBM graphs. The color bars attain the average costs, normalized by a factor of 1000, while the error bars indicate the max and min values, due to the randomness in graph generation and sampling Laplace noise. We can observe that our algorithm outperforms all other baselines by a considerable margin, especially when $\varepsilon$ is small. In addition, the cost of our algorithm is comparable to the non-private cost in some good runs. Note that SBM and HSBM graphs have well-clustered structures, therefore our algorithm has greater potential to perform well in practice when the input graph has an underlying clustered pattern.

**Real-world graphs.** Next, we evaluate our algorithm on real-world datasets. For each of the listed dataset, the similarity graph is constructed via the Gaussian kernel, where the parameters are chosen according to the standard heuristic (Ng et al., 2001), with the details in Appendix H. We use the same values for $\varepsilon$.

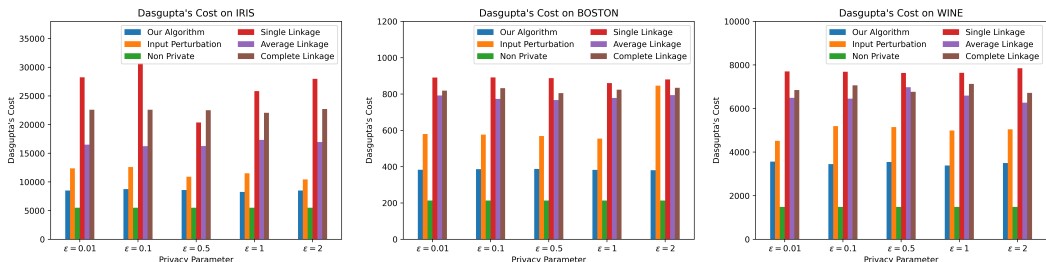

Figure 3: Comparison of Dasgupta's cost on real-world datasets: IRIS, WINE and BOSTON.

All results on different graphs are shown together in Figure 3. The reported cost takes the average of five runs. Similarly as the synthetic graphs, our algorithm outperforms the Input Perturbation method in terms of the additional cost by a large margin. Figure 1, Figure 2 and Figure 3 collectively show that our algorithm produces consistently better results across the range of $\varepsilon$, the privacy parameter.

An interesting observation is that contrary to many DP algorithms whose error is very sensitive with the choice of $\varepsilon$, the error does not change drastically with various values of $\varepsilon$. The reason is that, unlike conventional DP algorithms that release *values*, our algorithm releases the *HC trees*. The final cost is obtained by cost evaluation on the original graph. For the privately computed HC tree, even if we use a smaller $\varepsilon$ value, the *combinatorial structure* of the weights in the graph is relatively preserved by our algorithm. Therefore, the approximation error is less sensitive with the choice of $\varepsilon$.

## 4.2 Performance on Scalability

Scalability is a major gap between theory and practice on the performance of DP algorithms. Here we compare the running time of our algorithm, input perturbation and the algorithm for general graphs proposed in (Imola et al., 2023) with our own implementation. Note that the results in (Imola et al., 2023) are targeting SBM graphs, hence not applicable to general graphs. In fact, the algorithm for general graphs propopsed by (Imola et al., 2023) has exponential running time due to the exponential mechanism, therefore may break down even for $n > 10$.

Table 1: Comparison of Runtime (s) with $n = 6, 8, 10$

| HC Algorithm | $n = 6$ | $n = 8$ | $n = 10$ |
|---|---|---|---|
| HIERARCHICALCLUSTERING-DP (Our Algorithm) | 0.007 | 0.008 | 0.016 |
| Imola et al. (2023) (Our implementation) | 0.298 | 29.373 | >2h |
| Input Perturbation | 0.006 | 0.012 | 0.017 |
| Non-private Algorithm | 0.007 | 0.008 | 0.013 |

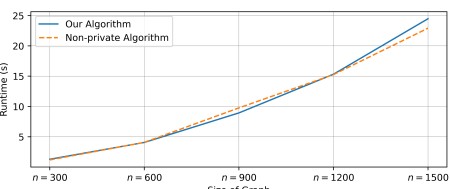

Figure 4: Runtime with $n$ scaling up to 1500

We first test all discussed HC algorithms on a cycle graph with $n = 6, 8, 10$. As shown in Table 1, the DP algorithm in (Imola et al., 2023) requires a huge amount of resources to handle a graph of size larger than 10. The rest three algorithms have relatively close running time. We further show in Figure 4 that our algorithm scales also similarly as the standard non-private algorithm. The graphs used in Figure 4 are generated from SBM with 5 clusters and varying sizes of each cluster.

# 5 A Lower Bound for the Balanced Sparsest Cut with Weight-level Privacy

We show the lower bound for the $\varepsilon$-DP balanced sparsest cut in the weight-level DP model as follows.

**Theorem 2** (Formalization of Result 3). *For any $\varepsilon \in (0, \frac{1}{20})$, any constant $\gamma$ and $n$ sufficiently large, any $\varepsilon$-weight DP algorithm for $\gamma$-balanced sparsest cut has to have $\Omega(\frac{1}{\varepsilon} \cdot \frac{1}{\log^2 n})$ expected additive error in sparsity, i.e., let $S$ be the output cut, let $\phi_G$ be the optimal sparsity, and let $\psi_G(S)$ be the sparsity of $S$, we have that*

$$\mathbb{E}\left[\psi_G(S)\right] > \phi_G + \Omega(\frac{1}{\varepsilon \log^2 n}).$$

Note that the output range of balanced sparsity can be as low as $\frac{1}{n}$ in even the unweighted graphs. As such, the $\Omega(\frac{1}{\varepsilon \log^2 n})$ is very significant in many instances. Furthermore, Theorem 2 implies a lower bound of $\Omega(\frac{n}{\varepsilon \log^2 n})$ additive error for balanced min-cuts (see Remark F.5 for details).

We prove Theorem 2 in this section. To begin with, we remind the readers of the notation: we use $\phi_G$ to denote the *minimum sparsity* of $G$, and we use $\psi_G(S)$ to denote the edge expansion of $S$ in $G$. We also focus on the case when $\gamma = \frac{1}{3}$, as the analysis for other constants follows from the same argument. Our main technical lemma for the proof is as follows.

**Lemma 5.1.** *Let $\mathcal{A}$ be an $\varepsilon$-DP algorithm that outputs a $\frac{1}{3}$-balanced sparsest cut with $\frac{C}{\varepsilon \log^2 n}$ expected additive error for some constant $C \in (0, 1)$. In other words, given a graph $G$, $\mathcal{A}$ outputs a partition of vertices $(S, V \setminus S)$ such that*

$$\mathbb{E}\left[\psi_G(S)\right] \le \phi_G + \frac{C}{\varepsilon \log^2 n}.$$

*Then, we can construct an $\varepsilon$-DP algorithm $\mathcal{A}'$ that outputs an HC tree $\mathcal{T}$, such that for any input graph $G = (V, E, w)$, there is*

$$\mathbb{E}\left[\mathsf{cost}_G(\mathcal{A}'(G))\right] \le O(1) \cdot \mathsf{OPT}_G + 2C \cdot \frac{n^2}{\varepsilon}$$

*for the same constant $C$ used by algorithm $\mathcal{A}$.*

Before showing the proof of Lemma 5.1, we first show how to use Lemma 5.1 to prove Theorem 2.

***Using Lemma 5.1 to prove Theorem 2.*** By Lemma 5.1, if there exists an algorithm $\mathcal{A}$ with $C \cdot \frac{1}{\varepsilon} \cdot \frac{1}{\log^2 n}$ expected additive error for $1/3$-balanced sparsest cut, it implies an HC algorithm $\mathcal{A}'$ with an expected cost of $O(1) \cdot \mathsf{OPT}_G + 2C \cdot \frac{n^2}{\varepsilon} < \mathsf{OPT}_G + \frac{1}{1200} \cdot \frac{n^2}{\varepsilon}$ for sufficiently small $C$. This forms a contradiction with our family of graphs as in Theorem 3 (on which the additive error should be at least $\frac{1}{1200} \cdot \frac{n^2}{\varepsilon}$), which means algorithm $\mathcal{A}$ for balanced sparsest cut cannot exist.   Theorem 2 □

**The reduction algorithm, $\mathcal{A}'$.**  To prove Lemma 5.1, we first show the reduction algorithm below.

---

**A reduction algorithm, $\mathcal{A}'$, from balanced sparsest cut to HC trees**

**Input:** A graph $G = (V, E, w)$; an algorithm, $\mathcal{A}$, that outputs a $\frac{1}{3}$-balanced sparsest cut with expected additive error at most $C \cdot \frac{1}{\varepsilon} \cdot \frac{1}{\log^2 n}$.

- Define *level $\ell$ cut* as the cut that happens with the distance $\ell - 1$ to the root of the HC tree (the entire graph $G$).
- Let $\mathcal{G}_\ell$ be the family of vertex-induced subgraphs in level $\ell$. Initialization $\mathcal{G}_1 = \{G\}$.
- For $\ell = 1 : \infty$
    1. For each vertex-induced subgraph $H \in \mathcal{G}_\ell$
        (a) If $|V(H)| = 1$, skip this set and go to the next $H \in \mathcal{G}_\ell$;
        (b) Otherwise, run $\mathcal{A}$ on $S$ with $\varepsilon_H = \frac{\varepsilon |H|}{10 n \log n}$, and obtain the partition of $H \to (S_1, S_2)$;
        (c) Add the partition $H \to (S_1, S_2)$ to the HC tree $\mathcal{T}$;
        (d) Add $G[S_1]$ and $G[S_2]$ to $\mathcal{G}_{\ell+1}$.
    2. Increase $\ell$ ($\ell \leftarrow \ell + 1$) and proceed to the next level.
    3. If all the vertex-induced subgraphs are singleton, terminate and output $\mathcal{T}$.

---

In other words, we run the recursive approximate balanced min-cut with $\mathcal{A}$, and as we go deeper down in the tree, we *decrease* the value of $\varepsilon$ adaptively. We defer the detailed analysis to Appendix F.

## 6 Conclusion and Discussion

In this paper, we revisited the cost of computing hierarchical clustering under the constraints of edge-level differential privacy. Previous work by Imola et al. (2023) showed that a lower bound of $\Omega(n^2/\varepsilon)$ (matched by exponential mechanism) and they gave an upper bound of $\widetilde{O}(n^{2.5}\sqrt{\log(1/\delta)}/\varepsilon)$ under $(\varepsilon, \delta)$-differential privacy. We argued that such additive error is too pessimistic for many natural classes of graphs. To assuage this concern, we explored the price of hierarchical clustering under weight-level differential privacy, where we showed that $\Omega(n^2/\varepsilon)$ additive error is unavoidable for any general graph, which is too high to be of any practical use.

In pursuit of understanding the least assumption on the input graph, we showed that could obtain a much more practical bound under a very mild (and practical) assumption. In particular, if the edge weights are at least 1, we can design a polynomial time $\varepsilon$-differentially private algorithm that achieves a purely multiplicative $\widetilde{O}(1/\varepsilon)$ approximation. Without privacy, getting an $\widetilde{O}(1)$ approximation is the ideal goal in the context of hierarchical clustering. Therefore, we show that there is little price of privacy for hierarchical clustering under an assumption that is satisfied by many downstream applications of hierarchical clustering. This is in contrast with what previous results suggested.

**Limitation and open problems.** Our positive results only provide a multiplicative approximation that is larger than the best possible approximation without privacy; however, it would be interesting to know if we can reduce the multiplicative approximation and understand potential tradeoffs between the multiplicative and additive approximation. Another open question to follow our work is HC with approximate privacy: with our privacy model, is it possible to get improved bounds if we focus on approximate DP instead?

ACKNOWLEDGEMENTS

Deng and Gao are supported by NSF IIS-2229876, DMS-2220271, DMS-2311064, CCF-2208663, CCF-2118953. Jalaj would like to acknowledge funding through Decanal Research Grant and unrestricted gift from Google. Samson Zhou is supported in part by NSF CCF-2335411. The work was conducted in part while Samson Zhou was visiting the Simons Institute for the Theory of Computing as part of the Sublinear Algorithms program.

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

# A    PRELIMINARIES AND KNOWN RESULTS

We introduce a few standard techniques in differential privacy and probability: Laplace mechanism, post-processing theorem, sum of Laplace random variables, composition theorem, etc.

**Definition 6** (Laplace distribution). We say a zero-mean random variable $X$ follows the Laplace distribution with parameter $b$ (denoted by $X \sim \mathsf{Lap}(b)$) if the probability density function of $X$ follows

$$p(x) = \mathsf{Lap}(b)(x) = \frac{1}{2b} \cdot \exp\left(-\frac{|x|}{b}\right).$$

**Definition 7** (Sensitivity). Let $p \geq 1$. For any function $f : \mathcal{X} \to \mathbb{R}^k$ defined over a domain space $\mathcal{X}$, the $\ell_p$-sensitivity of the function $f$ is defined as

$$\Delta_{f,p} = \max_{\substack{w,w' \in \mathcal{X} \\ w \sim w'}} \|f(w) - f(w')\|_p,$$

Here, $\|x\|_p := \left(\sum_{i=1}^d |x[i]|^p\right)^{1/p}$ is the $\ell_p$-norm of the vector $x \in \mathbb{R}^d$.

Based on Laplace distribution, we can now define Laplace mechanism – a standard DP mechanism that adds noise sampled from Laplace distribution with scale dependent on the $\ell_1$-sensitivity of the function. The formal definition is as follows.

**Definition 8** (Laplace mechanism). For any function $f : \mathcal{X} \to \mathbb{R}^k$, the Laplace mechanism on input $w \in \mathcal{X}$ samples $Y_1, \ldots, Y_k$ independently from $\mathsf{Lap}(\frac{\Delta_{f,1}}{\varepsilon})$ and outputs

$$M_\varepsilon(f) = f(w) + (Y_1, \ldots, Y_k).$$

**Proposition A.1** (Laplace mechanism Dwork et al. (2016)). *The Laplace mechanism $M_\varepsilon(f)$ is $\varepsilon$-differentially private.*

**Proposition A.2** (Post-processing theorem Dwork & Roth (2014)). *Let $M : \mathbb{R}^{d_1} \to \mathbb{R}^{d_2}$ be an $(\varepsilon, \delta)$-differentially private mechanism and let $g : \mathbb{R}^{d_2} \to \mathbb{R}^{d_3}$ be an arbitrary function. Then, the function $g \circ M : \mathbb{R}^{d_1} \to \mathbb{R}^{d_3}$ is also $(\varepsilon, \delta)$-differentially private.*

**Proposition A.3** (Sum of Laplace random variables, Chan et al. (2011); Wainwright (2019)). *Let $\{X_i\}_{i=1}^m$ be a collection of independent random variables such that $X_i \sim \mathsf{Lap}(b_i)$ for all $1 \leq i \leq m$. Then, for $\nu \geq \sqrt{\sum_i b_i^2}$ and $0 < \lambda < \frac{2\sqrt{2}\nu^2}{b}$ for $b = \max_i \{b_i\}$,*

$$\Pr\left[\left|\sum_i X_i\right| \geq \lambda\right] \leq 2 \cdot \exp\left(-\frac{\lambda^2}{8\nu^2}\right).$$

The following propositions characterize the *composition* of multiple differentially private algorithms.

**Proposition A.4** (Composition theorem Dwork et al. (2006; 2016); Rogers et al. (2016)). *Let $\{\mathcal{A}_j\}_{j=1}^k$ be $k$ algorithms with $(\varepsilon_j, 0)$-DP guarantees for $j \in [k]$. Furthermore, let $\mathcal{A}$ be a function of the outputs of $\mathcal{A}_1, \mathcal{A}_2, \cdots, \mathcal{A}_k$ with possible* dependent *calls and* adaptively chosen *parameters. Then, $\mathcal{A}$ is a $(\sum_{j=1}^k \varepsilon_j, 0)$-DP algorithm.*

**Proposition A.5** (Strong composition theorem Dwork et al. (2010)). *For any $\varepsilon, \delta \geq 0$ and $\delta' > 0$, the adaptive composition of $k$ $(\varepsilon, \delta)$-differentially private algorithms is $(\varepsilon', k\delta + \delta')$-differentially private for*

$$\varepsilon' = \sqrt{2k \ln(1/\delta')} \cdot \varepsilon + k\varepsilon(e^\varepsilon - 1).$$

*Furthermore, if $\varepsilon' \in (0,1)$ and $\delta' > 0$, the composition of $k$ $\varepsilon$-differentially private mechanism is $(\varepsilon', \delta')$-differentially private for*

$$\varepsilon' = \varepsilon \cdot \sqrt{8k \log(\frac{1}{\delta'})}.$$

We now move on to the definition of sparsest cut.

**Definition 9** (Sparsest Cut). *Given a weighted graph $G = (V, E, w)$ with $|V| = n$, let $S \subset V$ be a subset of vertices and $|S| \leq n/2$. Then the cut induced by $(S^*, \bar{S}^*)$ such that $S^* = \arg\min_{S \subseteq V} \frac{w(S, \bar{S})}{|S|}$ is the sparsest cut of $G$, where $\bar{S} = V \setminus S$.*

The optimal value of $\phi_G = \frac{w(S^*, \bar{S}^*)}{|S^*|}$ is also known as the *sparsity* of the graph. For the convenience of notation, we also use $\psi_G(S) = \frac{w(S, \bar{S})}{|S|}$ to denote the sparsity of $S$ in $G$, also known as edge expansion. We say the output $S$ is an $(\alpha, \beta)$-*approximation* of the sparsest cut if $\frac{w(S, \bar{S})}{|S|} \leq \alpha \cdot \phi_G + \beta$. Further, if $|S| = \gamma n$ for some $\gamma \in [\frac{1}{n}, \frac{1}{2}]$, the cut $(S, \bar{S})$ is called $\gamma$-*balanced sparsest cut*. Similarly, one can define $\gamma$-balanced min-cut by $S^* = \arg\min_{S \subseteq V} w(S, \bar{S})$ where $|S| = \gamma n$ for some $\gamma \in [\frac{1}{n}, \frac{1}{2}]$.

The statement below formally connects (balanced) sparsest cut with hierarchical clustering with Dasgupta's cost.

**Proposition A.6** (Charikar & Chatziafratis (2017); Dasgupta (2016)). *Let $\mathcal{T}$ be an HC tree that is obtained by recursively performing $\alpha$-approximate $1/3$-balanced sparsest cut on the vertex-induced subgraphs. Then, $\mathcal{T}$ gives an $O(\alpha)$-approximation to the optimal Dasgupta's HC cost.*

## B    ADDITIONAL RELATED WORK

We provide a discussion about additional related work in hierarchical clustering and similar DP graph problems.

**Hierarchical clustering with differential privacy.**    Obtaining differentially private algorithms for HC is a recent direction, and there are only a few works before Imola et al. (2023). To the best of our knowledge, Xiao et al. (2014) was the first work to investigate HC with privacy constraints. However, their work does *not* consider the optimization of objective functions. Kolluri et al. (2021) studied the differentially private algorithms under a maximization variant of Dasgupta's objective, and their results are not directly comparable with ours (or those of Imola et al. (2023)). Outside the regime of Dasgupta's objective, the Moseley-Wang objective (Moseley & Wang (2023)) is considered as the dual of Dasgupta's objective. Although there is no dedicated work for DP algorithms for the Moseley-Wang objective, it is known that recursive random cuts result in $1/3$-approximation for the objective, which, in turn, is also a DP algorithm due to the independence of the graph topology and weights.

**Minimum spanning trees with differential privacy.**    Another closely related graph problem that has been studied through the lens of privacy is the minimum spanning trees (MST). In this problem, the goal is to release a set of edges that form a tree and span all vertices. For this problem, a line of work by Sealfon (2016); Pinot (2018); Hladík & Teték (2024); Pagh & Retschmeier (2024) have shown that the tight error bound for the weight-neighboring case is $\Theta(n \log n/\varepsilon)$ for $\varepsilon$-DP algorithms. Hladík & Teték (2024); Pagh & Retschmeier (2024) further considered the aspect of universal optimality and running time. Moreover, these works also considered $\varepsilon$-DP algorithms for neighboring graphs with $\ell_\infty$ norm of the weights differs by at most 1, and obtained tight additive error bounds of $\Theta(n^2 \log n/\varepsilon)$. We note that the results on MST cannot be directly compared to our results: on a high level, the error induced by privacy mechanisms for the MST problem is easier to handle since the noises across edges do not accumulate.

## C    A LOWER BOUND FOR HIERARCHICAL CLUSTERING WITH WEIGHT-DP

Weight-differential privacy is a different notion than edge-differential privacy. Thus, it is not clear how the privacy-utility trade-off for hierarchical clustering differs across two privacy models (e.g. Chen et al. (2023); Sealfon (2016)). Specifically, the lower bound of $\Omega(n^2/\varepsilon)$ for the edge-level DP does not rule out a better additive error for the weight-level DP model. In this section, we show that the $\Omega(n^2/\varepsilon)$ lower bound can be extended to the weight-level DP model.

**Theorem 3** (Formalization of Result 2). *For any $\varepsilon \in (0, \frac{1}{20})$ and $n$ sufficiently large, there exists a graph $G$ of size $n$ such that the optimal HC cost is $O(n/\varepsilon)$, and any $\varepsilon$-weight DP algorithm for hierarchical clustering must have HC cost $\Omega(n^2/\varepsilon)$.*

**Our hard family of graphs.** We now construct our family of the hard instances. The starting point of our lower bound is a similar construction in Imola et al. (2023). In particular, Imola et al. (2023) used a hard family of instances as *random disjoint 5 cycles*. Since we operate with the weight-level DP, the graph topology is known, and we can no longer use the disjointness in the 5 cycles. Nevertheless, we can still exploit this structural property by putting extremely small weights on all edges that are *not* part of the 5 cycles. In this way, a low-cost algorithm still has to avoid cutting any cycle edges on the first levels of the HC tree – a property that is in tension with the privacy guarantee. The description of the hard family is as follows.

---

$\mathcal{D}(n, \varepsilon)$ : A hard family of graphs for hierarchical clustering with weight-level DP

- **Topology:** Let $G = (V, E)$ be a complete graph.
- **Edge weights:**
    1. Sample a collection of 5 cycles uniformly at random, and let this subgraph be $C_5$.
    2. For each $(u, v) \in E(C_5)$, let $w(u, v) = W$ with $W = \frac{1}{20} \cdot \frac{1}{\varepsilon}$.
    3. Otherwise, for each $(u, v) \notin E(C_5)$, let $w(u, v) = \frac{1}{n^3}$.

---

For the clarity of presentation, we first formally define the notions of the family and the collection of the trees with cost at most $r$ for each graph $G$ therein.

**Definition 10** (Imola et al. Imola et al. (2023), rephrased). Let $\mathcal{G}_{n,5}$ be the family of random disjoint 5-cycles where edges carry weights of 1. For each $G \in \mathcal{G}_{n,5}$, we define $\mathcal{B}(G, r) := \{\mathcal{T} \mid \mathsf{cost}_G(\mathcal{T}) \leq r\}$ be the family of HC trees such that the induced cost for $\mathcal{T}$ on $G$ is at most $r$. We call $\mathcal{B}(G, r)$ the collection of the trees of cost at most $r$ for $G$.

A key technical lemma of Imola et al. Imola et al. (2023) is that, if an algorithm is $\varepsilon$-DP, then the HC algorithm has to "incorrectly" cut many edges in the first partition. As a result, for many instances in the family of 5-cycles, their collections of trees of cost at most $O(n^2)$ are *disjoint*.

**Proposition C.1** (Imola et al. Imola et al. (2023), rephrased). *There exists a family of graphs $\mathcal{F}_{n,5} \subseteq \mathcal{G}_{n,5}$ such that $|\mathcal{F}_{n,5}| \geq 2^{n/5}$ and for any $G, G' \in \mathcal{F}_{n,5}$, $\mathcal{B}(G, \frac{n^2}{400}) \cap \mathcal{B}(G', \frac{n^2}{400}) = \emptyset$.*

Recall that in our hard family of $\mathcal{D}(n, \varepsilon)$, we have $\varepsilon < 1/20$ in Theorem 3, we always have $w(u, v) > 1$ for edges $(u, v) \in E(C_5)$. We also prove that the optimal HC cost of $G$ is at most $O(\frac{n}{\varepsilon})$.

**Lemma C.1.** *Fix any instance $G$ obtained by $\mathcal{D}(n, \varepsilon)$. The optimal HC cost of $G$ is at most $\frac{n}{\varepsilon}$.*

*Proof.* Recall that we use $\mathcal{T}^*$ to denote the optimal tree and $\mathsf{OPT}_G$ to denote the optimal cost of Dasgupta's HC on $G$. Let us construct a tree $\mathcal{T}$ such that $\mathsf{cost}_G(\mathcal{T}) \leq C \cdot \frac{n}{\varepsilon}$; and by definition of the optimal tree, we will have $\mathsf{OPT}_G \leq \mathsf{cost}_G(\mathcal{T}) \leq C \cdot \frac{n}{\varepsilon}$.

The construction of the tree $\mathcal{T}$ is simple: we keep dividing the vertices into groups of *disjoint* 5-cycles until the internal node contains 5 vertices. If an internal node $x$ contains exactly 5 vertices, we construct the subtree induced by $x$ by removing one vertex at each level. In this way, we can have a valid binary HC tree $\mathcal{T}$.

We can now partition the HC cost of $\mathcal{T}$ as follows.

$$\mathsf{cost}_G(\mathcal{T}) = \sum_{(u,v) \in E(C_5)} w(u, v) \cdot |\mathcal{T}[u \vee v]| + \sum_{(u,v) \notin E(C_5)} w(u, v) \cdot |\mathcal{T}[u \vee v]|.$$

Note that for each 5 cycle, we pay a cost of at most $W \cdot (10 + 4 + 3 + 2) = 19W$. Therefore, summing over all the $n/5$ such cycles, we have that

$$\sum_{(u,v) \in E(C_5)} w(u, v) \cdot |\mathcal{T}[u \vee v]| \leq \frac{n}{5} \cdot 19W = \frac{19n}{5} \cdot W.$$

On the other hand, note that each edge $(u, v)$ can induce a cost of at most $w(u, v) \cdot n$. Therefore, summing up at most $n^2/2$ such $(u, v)$ edges, we have that

$$\sum_{(u,v) \notin E(C_5)} w(u, v) \cdot |\mathcal{T}[u \vee v]| \leq \frac{n^2}{2} \cdot n \cdot \frac{1}{n^3} = \frac{1}{2}.$$

Using $W = \frac{1}{20\varepsilon}$ and $\varepsilon < \frac{1}{20}$, we have $\mathsf{cost}_G(\mathcal{T}) \leq \frac{19n}{5} \cdot W + \frac{1}{2} \leq O(\frac{n}{\varepsilon})$, as desired. $\qquad \square$

We now show a technical lemma that would "reduce" the collection of the low-cost trees for $G$ in the weight-neighboring model to the edge-neighboring model.

**Lemma C.2.** *Fix any $\varepsilon < 1$. Let $C_5, C'_5 \in \mathcal{G}_{n,5}$ be two random disjoint $5$-cycles, and let $\tilde{C}_5$ and $\tilde{C}'_5$ be the cycles with weight on each cycle $\frac{1}{20\varepsilon}$. Let $G, G'$ be two graphs obtained by $\mathcal{D}(n, \varepsilon)$ from $C_5$ and $C'_5$. If $\mathcal{B}(G, \frac{n^2}{400} \cdot W) \cap \mathcal{B}(G', \frac{n^2}{400} \cdot W) \neq \emptyset$, then $\mathcal{B}(\tilde{C}_5, \frac{n^2}{400} \cdot W) \cap \mathcal{B}(\tilde{C}'_5, \frac{n^2}{400} \cdot W) \neq \emptyset$.*

*Proof.* Recall that, for a graph $G$ and $r > 0$, $\mathcal{B}(G, r) := \{\mathcal{T} \mid \mathsf{cost}_G(\mathcal{T}) \leq r\}$ is the family of HC trees $\mathcal{T}$ such that the induced cost for $\mathcal{T}$ on $G$ is at most $r$. We prove the lemma by showing that for any HC tree $\mathcal{T}$, if $\mathcal{T} \in \mathcal{B}(G, \frac{n^2}{400} \cdot W)$, then $\mathcal{T} \in \mathcal{B}(\tilde{C}_5, \frac{n^2}{400} \cdot W)$. Note that this will give us $\mathcal{B}(G, \frac{n^2}{400} \cdot W) \subseteq \mathcal{B}(\tilde{C}_5, \frac{n^2}{400} \cdot W)$ (and resp. $\mathcal{B}(G', \frac{n^2}{400} \cdot W) \subseteq \mathcal{B}(\tilde{C}'_5, \frac{n^2}{400} \cdot W)$), which will lead to the desired result.

Fix an HC tree $\mathcal{T} \in \mathcal{B}(G, \frac{n^2}{400} \cdot W)$. We partition the HC cost and lower bound it with $\mathsf{cost}_{\tilde{C}_5}(\mathcal{T})$:

$$\mathsf{cost}_G(\mathcal{T}) = \sum_{(u,v) \in E(C_5)} w(u, v) \cdot |\mathcal{T}[u \vee v]| + \sum_{(u,v) \notin E(C_5)} w(u, v) \cdot |\mathcal{T}[u \vee v]|$$

$$= \mathsf{cost}_{\tilde{C}_5}(\mathcal{T}) + \sum_{(u,v) \notin E(C_5)} w(u, v) \cdot |\mathcal{T}[u \vee v]| \geq \mathsf{cost}_{\tilde{C}_5}(\mathcal{T}),$$

where the last equality is because $w(u, v) > 0$ in $G$. Since $\mathcal{T} \in \mathcal{B}(G, \frac{n^2}{400} \cdot W)$, we have that $\mathsf{cost}_G(\mathcal{T}) \leq \frac{n^2}{400} \cdot W$. This implies that $\mathsf{cost}_{\tilde{C}_5}(\mathcal{T}) \leq \mathsf{cost}_G(\mathcal{T}) \leq \frac{n^2}{400} \cdot W$ as desired. $\qquad \square$

We are now ready to prove Theorem 3 for the weight-level DP lower bound.

***Proof of Theorem 3.*** The proof basically follows the standard packing argument as in Imola et al. Imola et al. (2023) once we have established Lemma C.2; we provide a detailed proof for the purpose of completeness. Define $\mathcal{F}_{K_n}$ as the family of graphs in $\mathcal{D}(n, \varepsilon)$ such that the corresponding $5$-cycles are from $\mathcal{F}_{n,5}$. Note that in the neighboring graphs $G, G' \in \mathcal{F}_{K_n}$, the change of weights (in $L_1$ metric) are at most $2Wn$. Therefore, we can use the same group privacy argument as in Imola et al. Imola et al. (2023) to obtain an algorithm $\mathcal{A}$ that is $2\varepsilon Wn$-private on the graphs in the family $\mathcal{F}_{K_n}$. As such, fix any $G, G' \in \mathcal{F}_{K_n}$. Then

$$\Pr\left(\mathcal{A}(G) \in \mathcal{B}(G', \frac{n^2}{400} \cdot W)\right) \geq \exp\left(-2\varepsilon Wn\right) \cdot \Pr\left(\mathcal{A}(G') \in \mathcal{B}(G', \frac{n^2}{400} \cdot W)\right).$$

Suppose, for the sake of contradiction; there is an algorithm $\mathcal{A}$ which has an expected cost at most $\frac{n^2 \cdot W}{1200}$. That is, for $\mathbb{E}\left[\mathsf{cost}_{G'}(\mathcal{A}(G'))\right] \leq \frac{n^2 \cdot W}{1200}$. Markov inequality then implies that $\Pr(\mathsf{cost}_{G'}(\mathcal{A}(G')) \geq \frac{n^2 \cdot W}{400}) \leq \frac{1}{3}$. Therefore, we have that

$$\Pr\left(\mathcal{A}(G) \in \mathcal{B}(G', \frac{n^2}{400} \cdot W)\right) \geq \exp\left(-2\varepsilon Wn\right) \cdot \frac{2}{3} > \frac{1}{2^{n/5}}.$$

On the other hand, by the contrapositive of Lemma C.2, we know that if $\mathcal{B}(\tilde{C}_5, \frac{n^2}{400} \cdot W) \cap \mathcal{B}(\tilde{C}'_5, \frac{n^2}{400} \cdot W) = \emptyset$, there is $\mathcal{B}(G, \frac{n^2}{400} \cdot W) \cap \mathcal{B}(G', \frac{n^2}{400} \cdot W) = \emptyset$. As such, we can get that $|\mathcal{F}_{K_n}| \geq 2^{n/5}$, and for every pair of graph $G, G'$, $\mathcal{B}(G, \frac{n^2}{400} \cdot W) \cap \mathcal{B}(G', \frac{n^2}{400} \cdot W) = \emptyset$. Therefore, we can apply the standard union-bound argument and show the contradiction argument as follows.

$$1 = \sum_{G' \in \mathcal{F}_{K_n}} \Pr\left(\mathcal{A}(G) \in \mathcal{B}(G', \frac{n^2}{400} \cdot W)\right) \qquad \text{(by disjointness of the balls)}$$

$$> \sum_{G' \in \mathcal{F}_{K_n}} \frac{1}{2^{n/5}} \geq 2^{n/5} \cdot \frac{1}{2^{n/5}} = 1.$$

As such, we must have that $\mathbb{E}\left[\mathsf{cost}_{G'}(\mathcal{A}(G'))\right] > \frac{n^2 \cdot W}{1200} = \Omega(\frac{n^2}{\varepsilon})$ proving Theorem 3. □

# D    PROOFS FOR THE ALGORITHM IN SECTION 3

## D.1    PROOF OF LEMMA 3.1

*Proof.* The privacy guarantee follows from Step 2, where we are applying input perturbation. Each independent edge weight has sensitivity 1 corresponding to data publishing, therefore adding noise of $\mathsf{Lap}(1/\varepsilon)$ suffices for $\varepsilon$-DP (Proposition A.1). □

## D.2    PROOF OF LEMMA 3.2

*Proof.* First note that the SDP algorithm runs on graphs with positive edge weights. By the following claim we show this can be satisfied in BALANCEDSPARSESTCUT-DP with high probability.

**Claim D.1.** *For $G'' = (V, E, w'')$ defined in* BALANCEDSPARSESTCUT-DP*, with high probability, it holds for any edge $e \in E$ that $w''(e) \geq 0$.*

*Proof.* Let $X \sim \mathsf{Lap}(1/\varepsilon)$. Then we have:
$$\Pr(X \leq -10 \log n/\varepsilon) \leq e^{-10 \log n} = n^{-10}$$
By the union bound, all edge weights are non-negative with high probability.                    Claim D.1 □

We now continue with the proof. The multiplicative factor of $O(\log^{1.5} n/\varepsilon)$ comes in two parts. The algorithm in Proposition 3.1 contributes an $O(\sqrt{\log n})$ factor, thus it suffices to show that the perturbation in step 2 leads to $O(\log n/\varepsilon)$-approximation. Suppose the optimal cut of $G$ is $S$ such that $\phi_G = \frac{w(S, \bar{S})}{|S|}$. Let $S''$ be the optimal sparsest cut on $G''$. We have

$$\frac{w_{G''}(S'', \bar{S}'')}{|S''|} \leq \frac{w_{G''}(S, \bar{S})}{|S|} \qquad \text{(by the definition of sparsest cut of } G'')$$

$$= \frac{w_{G'}(S, \bar{S}) + \sum_{i=1}^{k} X_i}{|S|} \qquad \text{(by our construction)}$$

$$= \frac{w_G(S, \bar{S}) + \frac{10k \log n}{\varepsilon} + \sum_{i=1}^{k} X_i}{|S|} = \phi_G + \frac{\frac{10k \log n}{\varepsilon} + \sum_{i=1}^{k} X_i}{|S|},$$

where $k$ is the number of edges involved in the cut $(S, \bar{S})$, and $\{X_i\}_{i=1}^{k}$ are independent random variables such that $X_i \sim \mathsf{Lap}(1/\varepsilon)$. Recall the definition of our privacy model, we have $w(e) \geq 1$ for $\forall e \in E$, then $k \leq w_G(S, \bar{S})$. Applying Proposition A.3, we have:

$$\Pr\left(\left|\sum_{i=1}^{k} X_i\right| \geq \sqrt{80k \log n}/\varepsilon\right) \leq 2 \cdot \exp\left(-\frac{80k \log n/\varepsilon^2 \cdot \varepsilon^2}{8k}\right) = 2 \cdot n^{-10}.$$

That is, $\frac{1}{|S|} \sum_{i=1}^{k} X_i < \frac{10\sqrt{w(S, \bar{S}) \log n}}{\varepsilon |S|} < \frac{10 \log n}{\varepsilon} \cdot \phi_G$ holds with high probability. It follows that:

$$\frac{w_{G''}(S'', \bar{S}'')}{|S''|} \leq \phi_G + \frac{\frac{10k \log n}{\varepsilon} + \sum_{i=1}^{k} X_i}{|S|} < \phi_G + \frac{20\phi_G \log n}{\varepsilon} = O(\log n/\varepsilon) \cdot \phi_G,$$

as desired.                                                                                                Lemma 3.2 □

# E    MORE DISCUSSIONS ON THE APPROXIMATION FOR THEOREM 1

We discuss other forms of approximation for the algorithm in Theorem 1. In Section 1, we discussed that compared to additive errors, the *multiplicative* approximation better captures clusterability for hierarchical clustering. Nevertheless, we provide some analysis for the additive error of our algorithms.

### E.1 BLENDING ALGORITHMS WITH MULTIPLICATIVE AND ADDITIVE ERRORS

Note that in our privacy models, the algorithms in Imola et al. (2023) remain private. When $\mathsf{OPT}_G = o(n^2/\log^{0.5} n)$ ($\mathsf{OPT}_G = o(n^{2.5} \cdot \log^{0.5} n)$ for poly-time algorithms), our algorithm is strictly better than the algorithm in Imola et al. (2023). On the other hand, the algorithms in Imola et al. (2023) become more competitive when $\mathsf{OPT}_G$ becomes large. As such, a natural question is whether we could "adjust to" the algorithms based on the input instance and output with the one with the smaller error.

Running different algorithms in parallel and outputting the one with the smaller error is a standard trick in approximation algorithms. Nevertheless, for *differentially private* algorithm, we need to make sure that the selection between different algorithms retains privacy. For our purpose, even if we obtain an HC tree $\mathcal{T}$ from a private algorithm, some privacy loss will incur after we *evaluate* the HC cost on the *original graph* $G$. As such, we would need an $\varepsilon$-DP algorithm that could return the *cost* of an HC tree.

We present such an algorithm in this section and show that we could indeed get a smaller error among the additive and multiplicative algorithms (albeit with a small loss). Our main lemma for the private evaluation of the HC cost is as follows.

**Lemma E.1.** *There exist $\varepsilon$-weight DP algorithms such that given a weighted, connected graph $G$ of size $n$ with weight on each edge at least 1 and an HC tree $\mathcal{T}$, with high probability, in polynomial time computes a number $\widetilde{\mathsf{cost}}_G(\mathcal{T})$ such that*

$$\mathsf{cost}_G(\mathcal{T}) \leq \widetilde{\mathsf{cost}}_G(\mathcal{T}) \leq \mathsf{cost}_G(\mathcal{T}) + O(\frac{n \log n}{\varepsilon}),$$

*where $\mathsf{cost}_G(\mathcal{T})$ is the true (Dasgupta's HC objective) cost of $\mathcal{T}$ on $G$.*

*Proof.* The algorithm is simply as follows.

---
- Compute the cost of tree $\mathcal{T}$ on $G$.

- Add Laplace noise with $\mathsf{Lap}(\frac{n}{\varepsilon})$ to the output cost.
---

For privacy, we claim that the sensitivity of the function $\mathsf{cost}_G(\mathcal{T})$ is at most 1 for weight-neighboring graphs. Such a claim has been shown in Imola et al. (2023) (for the edge-neighboring model), and we give the statement and proof for the purpose of completeness.

**Claim E.2.** *Let $G = (V, E)$ be a graph and $w$ and $w'$ be neighboring weights as prescribed by Definition 3. Then, for any fixed HC tree $\mathcal{T}$, there is*

$$|\mathsf{cost}_{G,w}(\mathcal{T}) - \mathsf{cost}_{G,w'}(\mathcal{T})| \leq n,$$

*where $\mathsf{cost}_{G,w}(\cdot)$ and $\mathsf{cost}_{G,w'}(\cdot)$ are the (Dasgupta's HC objective) costs of $\mathcal{T}$ under $w$ and $w'$, respectively.*

*Proof.* The difference in Dasgupta's costs can be bounded as follows.

$$
\begin{aligned}
|\mathsf{cost}_{G,w}(\mathcal{T}) - \mathsf{cost}_{G,w'}(\mathcal{T})| &= \left| \sum_{(u,v) \in E} w(u,v) \cdot |\mathcal{T}[u \vee v]| - \sum_{(u,v) \in E} w'(u,v) \cdot |\mathcal{T}[u \vee v]| \right| \\
&= \left| \sum_{(u,v) \in E} (w(u,v) - w'(u,v)) \cdot |\mathcal{T}[u \vee v]| \right| \\
&\leq \sum_{(u,v) \in E} |w(u,v) - w'(u,v)| \cdot |\mathcal{T}[u \vee v]| \\
&\qquad\qquad\qquad (|\sum_i a_i| \leq \sum_i |a_i| \text{ for real numbers})
\end{aligned}
$$

$$\leq n \cdot \sum_{(u,v)\in E} |w(u,v) - w'(u,v)|$$

$$(|\mathcal{T}[u \vee v]| \leq n \text{ since the root contains at most } n \text{ vertices})$$

$$\leq n, \qquad \text{(by the weight-neighboring definition)}$$

as claimed. $\qquad\qquad\qquad\qquad\qquad\qquad\qquad\qquad\qquad\qquad$ Claim E.2 $\square$

As such, by Proposition A.1, the algorithm is $\varepsilon$-DP. Finally, for approximation guarantees, note that by the concentration of Laplace distribution, we have that

$$\Pr\left(\text{additive error} \geq 10 \cdot \frac{n \log n}{\varepsilon}\right) \leq \exp(-10 \log n) \leq \frac{1}{n^{10}},$$

which is as desired. $\qquad\qquad\qquad\qquad\qquad\qquad\qquad\qquad\qquad\qquad\qquad$ Lemma E.1 $\square$

Using Lemma E.1, we could perform the "outputting the algorithm with the smaller error", and obtain the following result.

**Lemma E.3.** *There exist a $(\varepsilon, \delta)$-weight DP algorithm such that given a weighted, connected graph $G$ of size $n$ with weight on each edge at least $1$, for any $\varepsilon \in (0,1)$, in polynomial time outputs an HC tree $\mathcal{T}$ whose cost is at most*

$$\min\left\{O\left(\frac{\log^{1.5} n \sqrt{\log(1/\delta)}}{\varepsilon}\right) \cdot \mathsf{OPT}_G + O\left(\frac{n \log n \sqrt{\log(1/\delta)}}{\varepsilon}\right), \; O(\sqrt{\log n}) \cdot \mathsf{OPT}_G + O\left(\frac{n^{2.5} \log^2 n \log^2(1/\delta)}{\varepsilon}\right)\right\},$$

*where $\mathsf{OPT}_G$ is the optimal HC cost of $G$ under Dasgupta's objective.*

*Proof.* The lemma follows by running both our algorithm and the algorithm in Imola et al. (2023), evaluating their cost privately with the algorithm of Lemma E.1, and outputting the HC tree with the smaller cost. Let $\mathcal{T}$ be the resulting HC tree from our algorithm and $\mathcal{T}'$ be the resulting HC tree from the algorithm in Imola et al. (2023). For the purpose of privacy, the parameter input for each algorithm is $\varepsilon' = O\left(\frac{\varepsilon}{\sqrt{\log(1/\delta)}}\right)$ (note that the only part we need to perform approximate DP is the algorithm of Imola et al. (2023)). Since the final algorithm is only a function of $\mathcal{T}, \mathcal{T}'$, $\widetilde{\mathsf{cost}}_G(\mathcal{T}), \widetilde{\mathsf{cost}}_G(\mathcal{T}')$, the privacy guarantee follows from the composition (Proposition A.5) and post-processing (Proposition A.2). Finally, the approximation guarantees follow from the respective error of the algorithms plus the additive error during the private cost evaluation as in Lemma E.1. $\square$

Note that the condition of $\varepsilon \in (0,1)$ is just for the conciseness of the statement. For $\varepsilon > 1$, we could similarly use the composition in Proposition A.5, albeit there will be an additional $\varepsilon \cdot (e^\varepsilon - 1)$ term. We omit the details for the cleanness of the presentation.

### E.2 EXPONENTIAL-TIME ALGORITHMS

We now show that if we are allowed to use exponential time, we can get stronger bounds for the multiplicative error than that of Theorem 1, and we can obtain an additive error of at most $O\left(\frac{n^2 \log n}{\varepsilon}\right)$ as in Imola et al. (2023). These algorithms are not practical; nonetheless, they highlight the extent of approximation we can get for $\varepsilon$-DP algorithms.

The formal statement is given as follows.

**Lemma E.4.** *There exist $\varepsilon$-weight DP algorithms such that given a weighted, connected graph $G$ of size $n$ with weight on each edge at least $1$, in exponential time outputs an HC tree $\mathcal{T}$ whose cost is at most*

$$\min\left\{O\left(\frac{\log n}{\varepsilon}\right) \cdot \mathsf{OPT}_G + O\left(\frac{n \log n}{\varepsilon}\right), \; \mathsf{OPT}_G + O\left(\frac{n^2 \log n}{\varepsilon}\right)\right\},$$

*where $\mathsf{OPT}_G$ is the optimal HC cost of $G$ under Dasgupta's objective.*

*Proof.* The algorithm could be obtained by the algorithm from Imola et al. Imola et al. (2023), our HIERARCHICALCLUSTERING-DP in Theorem 1, and the private evaluation algorithm of Lemma E.1. As such, we only sketch the proofs, and leave the full proof as an exercise for keen readers.

For the algorithm with $O(\frac{\log n}{\varepsilon})$ multiplicative error, we can run HIERARCHICALCLUSTERING-DP in the following manner: on step 3 of BALANCEDSPARSESTCUT-DP, instead of running the approximation algorithm in Proposition 3.1, we use $2^n$ time to enumerate all cuts, and output the cut $S$ with the lowest sparsity. With a minor modification of the analysis of Theorem 1, it is easy to see that the output cut is an $O(\frac{\log n}{\varepsilon})$ multiplicative approximation of the 1/3-balanced sparsest cut. As such, we can use Proposition A.6 to obtain the desired approximation guarantee for HC.

For the algorithm with $O(\frac{n^2 \log n}{\varepsilon})$ additive error, we can run the exponential mechanism as in Imola et al. Imola et al. (2023). Finally, with the same argument of Lemma E.3, we can run both algorithms and their private cost evaluation with parameter $\varepsilon/4$, and use composition (Proposition A.5) and post-processing (Proposition A.2) to obtain the desired statement. □

### E.3 APPROXIMATION GUARANTEES ON GRAPHS WITH < 1 WEIGHTS

In this section, we discuss the implication of our algorithm on graphs with weights *less than* 1. We show that the approximation guarantee of HIERARCHICALCLUSTERING-DP is fairly robust, and for any graph with the minimum weight at least $\alpha$, we could achieve $O(\frac{\log^{1.5} n}{\alpha \cdot \varepsilon})$-multiplicative approximation.

**Proposition E.1.** *The algorithm* HIERARCHICALCLUSTERING-DP *is $\varepsilon$-weight DP. Furthermore, given a weighted, connected graph $G$ of size $n$ with weight on each edge at least $\alpha$,* HIERARCHICALCLUSTERING-DP *outputs a HC tree $\mathcal{T}$ such that the HC cost by $\mathcal{T}$ is at most $O(\frac{\log^{1.5} n}{\alpha \cdot \varepsilon}) \cdot \mathsf{OPT}_G$, where $\mathsf{OPT}_G$ is the optimal HC cost of $G$ under Dasgupta's objective.*

Proving Proposition E.1 would require a repetition of all the steps we did in Theorem 1. Therefore, we provide a proof sketch on why the lemma is true. The privacy guarantee of Proposition E.1 follows directly from Lemma 3.1, and the approximation guarantee is based on the following lemma.

**Lemma E.5.** *With high probability, for any graph $G = (V, E, w)$ whose minimum weight is at least $\alpha$, the* BALANCEDSPARSESTCUT-DP *algorithm returns a balanced cut in polynomial-time with at most $O(\frac{\log^{1.5} n}{\alpha \cdot \varepsilon})$-multiplicative approximation to the sparsity of the balanced sparsest cut.*

*Proof.* The proof of Lemma E.5 follows by the same argument of the proof of Lemma 3.2 (as in Appendix D.2) with slight modifications. In particular, by the same argument, we let $S$ be the the *optimal* sparsest cut $S$ of $G$, and suppose there are $k$ edges in the cut. Let $X_i$ be the random variable for the Laplacian noise added on the $i$-th edge of the cut $(S, |S|)$, we could again show that

$$\frac{w_{G''}(S'', \bar{S}'')}{|S''|} \leq \phi_G + \frac{\frac{10k \log n}{\varepsilon} + \sum_{i=1}^{k} X_i}{|S|}.$$

Furthermore, by the argument with concentration inequalities, we have that

$$\frac{1}{|S|} \cdot \sum_{i=1}^{k} X_i < \frac{10\sqrt{k \cdot \log n}}{\varepsilon |S|}.$$

Note that by the condition of $w(e) \geq \alpha$ for all $e \in E$, we have that $k \leq \frac{w_G(S, \bar{S})}{\alpha}$. As such, we have that

$$\frac{w_{G''}(S'', \bar{S}'')}{|S''|} < \phi_G + 20\phi_G \log \frac{n}{\alpha \cdot \varepsilon} = O(\log \frac{n}{\alpha \cdot \varepsilon}) \cdot \phi_G.$$

Finally, as in the proof of Lemma 3.2, an extra factor of $O(\sqrt{\log n})$ is induced by Proposition 3.1, which gives the desired lemma statement. □

The multiplicative approximation bound of Proposition E.1 then follows from the same argument as the proof of Theorem 1 by losing a factor of $O(1/\alpha)$.

## E.4 $\varepsilon$-DP POLYNOMIAL-TIME ALGORITHM WITH $O(\frac{n^3 \log^{1.5} n}{\varepsilon})$ ADDITIVE ERROR

We now give a polynomial-time $\varepsilon$-DP algorithm that gives $O(\sqrt{n})$-multiplicative approximation with $O(\frac{n^3 \log^{1.5} n}{\varepsilon})$ additive error. We note that the approximation guarantee of this algorithm is strictly worse than the algorithm of Lemma E.3. Nevertheless, the algorithm is pure DP, and it does not need to call the algorithm in Imola et al. (2023) (note that it is unclear how the poly-time algorithm in Imola et al. (2023) could be implemented). As such, we believe the algorithm has its own advantage for certain scenarios.

**Lemma E.6.** *Given a weighted, connected graph $G$ of size $n$ with weight on each edge at least 1, for any $\varepsilon > 0$, the HC tree $\mathcal{T}$ computed by* HIERARCHICALCLUSTERING*-DP in Section 3.2 has cost at most* $\mathsf{cost}_G(\mathcal{T}) \leq O(\sqrt{\log n}) \cdot \mathsf{OPT}_G + O(\frac{n^3 \log^{1.5} n}{\varepsilon})$, *where* $\mathsf{OPT}_G$ *is the optimal HC cost of $G$ under Dasgupta's objective.*

The key notion we need to prove Lemma E.6 is the *optimal balanced tree* $\mathcal{T}^*_{\text{balance}}$. We now formally define the notion and analyze its properties.

**Definition 11.** We say an HC tree $\mathcal{T}_{\text{balance}}$ is a *balanced tree* if *every* internal node is a $1/3$-balanced partition, i.e., for any partition $H \to (S, H \setminus S)$ in $\mathcal{T}_{\text{balance}}$, there is $\min\{|S|, |H \setminus S|\} \geq \frac{|H|}{3}$.

We use $\mathcal{T}^*_{\text{balance}}$ to denote the optimal tree among the balanced tree, i.e., $\mathsf{cost}_G(\mathcal{T}^*_{\text{balance}}) \leq \mathsf{cost}_G(\mathcal{T}_{\text{balance}})$ for any balanced tree $\mathcal{T}_{\text{balance}}$.

By Proposition A.6, we can immediately observe that the tree $\mathcal{T}^*_{\text{balance}}$ achieves an $O(1)$-approximation to the optimal HC tree $\mathcal{T}^*$.

**Observation E.1.** *Let $\mathcal{T}^*_{balance}$ be the optimal tree among balanced trees on $G$. Then, we have*

$$\mathsf{cost}_G(\mathcal{T}^*_{balance}) \leq O(1) \cdot \mathsf{OPT}_G.$$

Observation E.1 is true simply because the tree we obtained by the recursive balanced sparsest cut is a balanced tree. We now use balanced trees as a 'bridge' between the costs on $G$ and the perturbed graph $G''$.

**Lemma E.7.** *Let $\mathcal{T}_{balance}$ be any balanced HC tree as defined by Definition 11, and let $G''$ be the graph obtained by steps 1 and 2 of* BALANCEDSPARSESTCUT*-DP. Then, with high probability, we have that*

$$\mathsf{cost}_G(\mathcal{T}_{balance}) \leq \mathsf{cost}_{G''}(\mathcal{T}_{balance}) \leq \mathsf{cost}_{G''}(\mathcal{T}_{balance}) + O(\frac{n^3 \log n}{\varepsilon}).$$

*Proof.* We prove the first inequality by showing that with high probability, $w''(u,v) \geq w(u,v)$. This is essentially the same proof as Claim D.1 – we can show that with probability at least $1 - 1/n^3$, there is $w''(u,v) \geq w(u,v)$ for all edges $(u,v) \in E$. Therefore, the cost of the same tree $\mathcal{T}_{\text{balance}}$ on $G''$ cannot be lower than the cost of $\mathcal{T}_{\text{balance}}$ on $G$.

For the second inequality, we first observe that by step 1, the additive error on each edge is $\frac{10 \log n}{\varepsilon}$. We now show that with high probability, the additive error on all edges by the Laplacian noise is at most $\frac{10 \log n}{\varepsilon}$. Again, this is by the same calculation as in Claim D.1, as follows. Let $X_{u,v} \sim \mathsf{Lap}(1/\varepsilon)$ be the random variable for the Laplacian noise on $(u,v)$, we have

$$\Pr(X_{u,v} \geq 10 \log n / \varepsilon) \leq e^{-10 \log n} = n^{-10},$$

and we can apply a union bound over all edges to get that with high probability, $w_{G''}(u,v) \leq w_G(u,v) + \frac{20 \log n}{\varepsilon}$ for all $(u,v) \in E$.

We now bound the additive error for $\mathcal{T}_{\text{balance}}$ from $G''$ to $G$. In the same way as the proof of Lemma 5.1, we define *level $\ell$ cut* as the cut that happens with the distance $\ell - 1$ to the root of the HC tree. Let $H \to (S, H \setminus S)$ be a partition in $\mathcal{T}_{\text{balance}}$ of level $\ell$, we first observe that

$$(\text{\# of edges in } \ell\text{-level partition } H \to (S, H \setminus S)) \leq (\frac{2}{3})^{2(\ell-1)} n^2$$

since bigger partition reduces size by at least a factor of $\frac{2}{3}$. Therefore, we can upper-bound the cost of $\mathcal{T}_{\text{balance}}$ in $G''$ as

$$\mathsf{cost}_{G''}(\mathcal{T}_{\text{balance}})$$

$$= \sum_{\substack{H \to (S, H \setminus S) \\ \text{induced by nodes in } \mathcal{T}}} w_{G''}(S, H \setminus S) \cdot |H|$$

$$= \sum_{\ell=1}^{\ell_{\max}} \sum_{\substack{H \to (S, H \setminus S) \\ \text{induced by nodes in } \mathcal{T} \\ \text{on level } \ell}} w_{G''}(S, H \setminus S) \cdot |H|$$

$$= \sum_{\ell=1}^{\ell_{\max}} \sum_{\substack{H \to (S, H \setminus S) \\ \text{induced by nodes in } \mathcal{T} \\ \text{on level } \ell}} \sum_{(u,v) \in E(S, H \setminus S)} w_{G''}(u, v) \cdot |H|$$

$$\leq \sum_{\ell=1}^{\ell_{\max}} \sum_{\substack{H \to (S, H \setminus S) \\ \text{induced by nodes in } \mathcal{T} \\ \text{on level } \ell}} \sum_{(u,v) \in E(S, H \setminus S)} \left( w_G(u, v) + \frac{20 \log n}{\varepsilon} \right) \cdot |H|$$

$$\text{(by the relationship between } w_G(u, v) \text{ and } w_{G''}(u, v))$$

$$\leq \sum_{\ell=1}^{\ell_{\max}} \sum_{\substack{H \to (S, H \setminus S) \\ \text{induced by nodes in } \mathcal{T} \\ \text{on level } \ell}} \left( w_G(S, H \setminus S) + \frac{20 \log n}{\varepsilon} \cdot (\frac{2}{3})^{2(\ell-1)} n^2 \right) \cdot |H|$$

$$\text{(by the bound of the number of edges in the partition)}$$

$$\leq \sum_{\ell=1}^{\ell_{\max}} \sum_{\substack{H \to (S, H \setminus S) \\ \text{induced by nodes in } \mathcal{T} \\ \text{on level } \ell}} w_G(S, H \setminus S) \cdot |H|$$

$$+ \frac{20 n^2 \log n}{\varepsilon} \cdot \sum_{\ell=1}^{\ell_{\max}} \sum_{\substack{H \to (S, H \setminus S) \\ \text{induced by nodes in } \mathcal{T} \\ \text{on level } \ell}} |H| \cdot (\frac{2}{3})^{2(\ell-1)}$$

$$\leq \mathsf{cost}_G(\mathcal{T}_{\text{balance}}) + \frac{20 n^2 \log n}{\varepsilon} \cdot n \cdot \sum_{\ell=1}^{\ell_{\max}} 2^{\ell-1} \cdot (\frac{2}{3})^{2(\ell-1)}$$

$$(|H| \leq n \text{ and there are } 2^{\ell-1} \text{ partitions on level } \ell)$$

$$\leq \mathsf{cost}_G(\mathcal{T}_{\text{balance}}) + O(\frac{n^3 \log n}{\varepsilon}), \qquad (\sum_{\ell=1}^{\infty} 2^{\ell-1} \cdot (\frac{2}{3})^{2\ell-1} = O(1))$$

as desired. $\qquad \square$

***Finalizing the proof of Lemma E.6.*** We run $O(\sqrt{\log n})$-approximate balanced sparsest cut on $G''$ in HIERARCHICALCLUSTERING-DP. In the end, the algorithm produces a tree $\mathcal{T}$ such that

$$\mathsf{cost}_{G''}(\mathcal{T}) \leq O(\sqrt{\log n}) \cdot \mathsf{cost}_{G''}(\mathcal{T}^*(G'')) \leq O(\sqrt{\log n}) \cdot \mathsf{cost}_{G''}(\mathcal{T}_{\text{balance}}^*),$$

where $\mathcal{T}^*(G'')$ is the optimal HC tree on $G''$. Since both $\mathcal{T}$ and $\mathcal{T}_{\text{balance}}^*$ are balanced trees, we have

$$\mathsf{cost}_G(\mathcal{T}) \leq \mathsf{cost}_{G''}(\mathcal{T}) \qquad \text{(using the first inequality of Lemma E.7)}$$

$$\leq O(\sqrt{\log n}) \cdot \mathsf{cost}_{G''}(\mathcal{T}_{\text{balance}}^*)$$

$$\leq O(\sqrt{\log n}) \cdot \left( \mathsf{cost}_G(\mathcal{T}_{\text{balance}}^*) + O(\frac{n^3 \log n}{\varepsilon}) \right)$$

$$\text{(using the second inequality of Lemma E.7)}$$

$$\leq O(\sqrt{\log n}) \cdot \mathsf{cost}_G(\mathcal{T}_{\text{balance}}^*) + O(\frac{n^3 \log^{1.5} n}{\varepsilon}),$$

as desired by the lemma statement. $\qquad$ Lemma E.6 $\square$

## F  THE ANALYSIS OF THE LOWER BOUND IN SECTION 5

In this section, we provide the analysis of the reduction for the balanced sparsest cut lower bound.

**High-level technical overview.  The balanced sparsest cut lower bound.** The connection between the balanced sparsest cut and hierarchical clustering is well-known in the literature. Therefore, a natural idea to prove lower bounds for $\varepsilon$-DP balanced sparsest cuts is through a reduction argument from $\varepsilon$-DP hierarchical clustering.

There are two technical challenges to formalizing the above idea. The first challenge is that existing results between balanced sparsest cuts and HC focus on *multiplicative* error, while we need to show a lower bound with *additive* error. This would require us to open the black box of the existing technical lemmas, and use a white-box adaptation of the "charging" argument of Charikar & Chatziafratis (2017). The second, and perhaps the bigger challenge, is the loss of privacy in the process. To obtain the HC tree, we need to run the $\varepsilon$-DP algorithm for the balanced sparsest cut on vertex-induced subgraphs, which requires repeated queries for up to $\Omega(n)$ times. As such, by the strong composition theorem, we need to lose a factor of $\sqrt{n}$ on the additive error, and we can only obtain an HC algorithm for $(\varepsilon, \delta)$-approximate DP – for which we can only prove lower bounds for very small $\delta$[3]!

Our idea to tackle this issue is to use more *restrictive* privacy parameters as we go down the HC tree with the balanced sparsest cuts. For the cuts on level $\ell$ (from the root), we run the DP balanced sparsest cut algorithm with $\varepsilon$ scaling with the *size* of the current partition, i.e., when we run the algorithm on $H$, we use $\varepsilon_H = \frac{|H|}{n} \cdot \varepsilon$ (we use $|H|$ for the number of vertices in $H$). Since we have $\sum_H \frac{|H|}{n} = 1$ on each level, the privacy loss on this particular level is at most $\varepsilon$. Furthermore, since the depth of the HC tree $\mathcal{T}$ can be at most $O(\log n)$ due to balanced cuts, the privacy loss is at most $O(\log n)$, which can be handled by rescaling.

The final missing piece here is the impact of changed $\varepsilon$ in the *additive error*. Since we decrease the parameter $\varepsilon$, the additive error increases. Nevertheless, we can show that on each partition, the blow-up is roughly $O(\frac{n|H|}{\varepsilon})$. Therefore, if we sum up the partitions of a level, the additive error is bounded by $O\left(\frac{n^2}{\varepsilon}\right)$. As such, we can again use the fact that the HC tree is balanced to obtain the total additive error of $O(\frac{n^2 \log n}{\varepsilon})$ – and the extra cost can again be handled by rescaling of $\varepsilon$.

**The privacy analysis.**  We first show that the reduction algorithm is $\varepsilon$-DP. The formal statement is as follows.

**Lemma F.1.** *The reduction algorithm that outputs the hierarchical clustering $\mathcal{T}$ satisfies $\varepsilon$-DP.*

*Proof.* First, note that we adaptively provide a privacy budget in Step 1(b); however, it should be noted that $H$ is computed using a private mechanism, so this choice in the reduction is privacy-preserving as long as the total privacy budget is at most $\varepsilon$. The final output HC tree $\mathcal{T}$ is a function only depending on the internal balanced sparsest cuts. In other words, $\mathcal{T}$ is a function of the algorithms $\{\mathcal{A}_j\}_{j=1}^k$, where $\mathcal{A}_j$ is the algorithm $\mathcal{A}$ induced on the internal node $j$, and $k$ is the total number of internal nodes. Let $\varepsilon(k)$ be the privacy parameter of the cut on internal node $k$. By Proposition A.4, the reduction algorithm satisfies DP with parameter

$$\sum_{j=1}^{k} \varepsilon(k) \leq \sum_{\ell=1}^{\ell_{\max}} \sum_{H \text{ on level } (\ell-1)} \varepsilon_H = \frac{1}{2 \log n} \cdot \ell_{\max} \cdot \varepsilon,$$

where $\ell_{\max}$ is the maximum level of the HC tree, and the last equation is due to $\sum_{H \text{ on level } (\ell-1)} \varepsilon_H = \varepsilon$ for every level $\ell$.

Since the tree is always $\frac{1}{3}$-balanced, the number of levels is at most $\log_{3/2} n \leq 2 \log n$ levels. As such, we have that the privacy parameters satisfies $\frac{1}{2 \log n} \cdot \ell_{\max} \cdot \varepsilon \leq \varepsilon$, as desired.  $\square$

---

[3]Although we did not include a proof for the HC lower bounds with $(\varepsilon, \delta)$-DP, it appears the proof in Appendix C would still work for $\delta = 1/2^{n/5}$.

**The utility analysis.** We now proceed to the utility analysis of the reduction algorithm. To this end, we first show that the balanced sparsest cut guarantee would lead to balanced min-cut guarantees, and the latter is easier to work with.

**Claim F.2.** *Let $(S, V \setminus S)$ be a $\frac{1}{3}$-balanced sparsest cut with $\frac{C}{\varepsilon \log^2 n}$ additive error for some constant $C \in (0, 1)$. Then, $S$ is an $\left(2, \frac{C}{2\varepsilon \log^2 n}\right)$-approximation of the $1/3$-balanced min-cut. In other words, let $(S^{min}, V \setminus S^{min})$ be the $1/3$-balanced min-cut of $G$, we have that*

$$w(S, V \setminus S) \leq 2 \cdot w(S^{min}, V \setminus S^{min}) + \frac{Cn}{2\varepsilon \log^2 n}.$$

*Proof.* Let $S^{sparse}$ be the $\frac{1}{3}$-balanced sparsest cut of $G$. By the guarantee of $\mathcal{A}$, we have that

$$
\begin{aligned}
w(S, V \setminus S) &\leq \frac{w(S^{sparse}, V \setminus S^{sparse})}{|S^{sparse}|} \cdot |S| + \frac{C}{\varepsilon \log^2 n} \cdot |S| && \text{(by the property of } \mathcal{A}) \\
&\leq \frac{w(S^{min}, V \setminus S^{min})}{|S^{min}|} \cdot |S| + \frac{C}{\varepsilon \log^2 n} \cdot |S| && \text{(by the definition of the sparsest cut)} \\
&\leq 2 \cdot w(S^{min}, V \setminus S^{min}) + \frac{Cn}{2\varepsilon \log^2 n}, && (\frac{|S|}{|S^{min}|} \leq 2 \text{ and } |S| \leq \frac{n}{2})
\end{aligned}
$$

as desired. $\square$

Let the output of $\mathcal{A}'$ on the input graph $G$ be $\mathcal{A}'(G) = \mathcal{T}$. Using the expression of the cost function as stated in eq (1), the expected cost of $\mathcal{T}$ is

$$
\begin{aligned}
\mathbb{E}\left[\text{cost}_G(\mathcal{T})\right] &= \mathbb{E}\left[\sum_{\substack{H \to (S_1, S_2) \\ \text{induced by nodes in } \mathcal{T}}} w(S_1, S_2) \cdot |H|\right] \\
&= \sum_{\substack{H \to (S_1, S_2) \\ \text{induced by nodes in } \mathcal{T}}} \mathbb{E}\left[w(S_1, S_2)\right] \cdot |H| && \text{(by linearity of expectation)} \\
&\leq \sum_{\substack{H \to (S_1, S_2) \\ \text{induced by nodes in } \mathcal{T}}} 2 \cdot w(S_H^{min}, V(H) \setminus S_H^{min}) \cdot |H| \\
&\qquad\qquad (S_H^{min} \text{ is the balanced min-cut for vertex-induced graph } H) \\
&\quad + \sum_{\ell=1}^{\ell_{max}} \sum_{\substack{H \to (S_1, S_2) \\ \text{induced by nodes in } \mathcal{T} \\ \text{on level } \ell}} C \cdot \frac{1}{2\varepsilon_H} \cdot \frac{|H|}{\log^2 n} \cdot |H|. \\
&\qquad\qquad \text{(by the property of the algorithm } \mathcal{A} \text{ and Claim F.2)}
\end{aligned}
$$

Note that in the last term, we have $\frac{|H|}{\log^2 n}$ as opposed to $\frac{n}{\log^2 n}$ since we are operating on the vertex-induced subgraph $H$.

We now analyze the two cost terms respectively. We first bound the summation of the former term.

**Lemma F.3.** *Let $\mathcal{T}$ be the tree obtained by recursive $1/3$-balanced cut whose cuts are represented by $H \to (S_1, S_2)$. Furthermore, let $(S_H^{min}, V(H) \setminus S_H^{min})$ be the balanced min-cuts of $V(H)$. Then, we have that*

$$\sum_{\substack{H \to (S_1, S_2) \\ \text{induced by nodes in } \mathcal{T}}} 2 \cdot w(S_H^{min}, V(H) \setminus S_H^{min}) \cdot |H| \leq O(1) \cdot \text{OPT}_G.$$

The proof of Lemma F.3 requires a white-box adaptation of the charging argument used by Charikar & Chatziafratis (2017) (see also, e.g. Agarwal et al. (2022); Assadi et al. (2022)). The idea is

almost identical to Charikar and Chatziafratis Charikar & Chatziafratis (2017), but the argument is considerably involved. As such, we postpone the proof to Appendix G.

We now analyze the summation

$$\sum_{\ell=1}^{\ell_{\max}} \sum_{\substack{H \to (S_1, S_2) \\ \text{induced by nodes in } \mathcal{T}}} C \cdot \frac{1}{2\varepsilon_H} \cdot \frac{|H|}{\log^2 n} \cdot |H|$$

for the additive error. The main lemma for this term is as follows.

**Lemma F.4.** *Let $\mathcal{T}$ be the tree obtained by recursive $1/3$-balanced cut whose cuts are represented by $H \to (S_1, S_2)$. Then, we have that*

$$\sum_{\ell=1}^{\ell_{max}} \sum_{\substack{H \to (S_1, S_2) \\ \text{induced by nodes in } \mathcal{T} \\ \text{on level } \ell}} C \cdot \frac{1}{2\varepsilon_H} \cdot \frac{|H|}{\log^2 n} \cdot |H| \leq C \cdot \frac{n^2}{\varepsilon}.$$

*Proof.* Note that at level $\ell$, the cost is at most

$$\sum_{H \text{ on level } (\ell - 1)} \frac{C}{2\varepsilon_H} \cdot \frac{|H|^2}{\log^2 n} \qquad s.t. \sum_{H \text{ on level } (\ell - 1)} |H| = n.$$

By plugging in our choice of $\varepsilon_H = \frac{1}{2 \log n} \cdot \varepsilon \cdot \frac{|H|}{n}$, we have that

$$\sum_{H \text{ on level } (\ell - 1)} \frac{C}{2\varepsilon_H} \cdot \frac{|H|^2}{\log^2 n} = \frac{C}{2\varepsilon} \cdot \sum_{H \text{ on level } (\ell - 1)} \left( \frac{2n \log n}{|H|} \cdot \frac{|H|^2}{\log^2 n} \right)$$

$$= \frac{Cn}{\varepsilon \log n} \cdot \sum_{H \text{ on level } (\ell - 1)} |H|$$

$$= \frac{Cn^2}{\varepsilon \log n}. \qquad \text{(since } \sum_{H \text{ on level } (\ell - 1)} |H| = n\text{)}$$

Finally, we get the tree $\mathcal{T}$ using $1/3$-balanced cuts. As such, there are at most $\log_{3/2} n \leq 2 \log n$ levels. Therefore, the total cost induced by the summation in Lemma F.4 is at most

$$\sum_{\ell=1}^{\ell_{max}} \sum_{\substack{H \to (S_1, S_2) \\ \text{induced by nodes in } \mathcal{T} \\ \text{on level } \ell}} \frac{C}{2\varepsilon_H} \cdot \frac{|H|}{\log^2 n} \cdot |H| \leq 2 \log n \cdot \frac{Cn^2}{\varepsilon \log n} = 2C \cdot \frac{n^2}{\varepsilon},$$

as desired. $\qquad \square$

**Wrapping up the proof of Lemma 5.1.** By Lemma F.1, the reduction algorithm is $\varepsilon$-DP. Furthermore, by Lemma F.3 and Lemma F.4, we know that

$$\mathbb{E}\left[\text{cost}_G(\mathcal{T})\right] \leq O(1) \cdot \text{OPT}_G + 2C \cdot \frac{n^2}{\varepsilon},$$

as desired by the lemma statement.

**Remark F.5.** *In the proof of Lemma 5.1, we essentially embedded a reduction from balanced sparsest cut to balanced min-cut. Using the same argument, we could argue that for balanced min-cut, the additive error for any $\varepsilon$-DP algorithm is at least $\Omega(\frac{n}{\varepsilon \log^2 n})$. The result essentially follows from Lemmas F.3 and F.4, and we omit the repetition to write the proof.*

# G    THE PROOF OF LEMMA F.3

We provide the proof of Lemma F.3 in this section. To begin with, we first recap the lemma statement as follows.

**Lemma F.3.** *Let $\mathcal{T}$ be the tree obtained by recursive $1/3$-balanced cut whose cuts are represented by $H \to (S_1, S_2)$. Furthermore, let $(S_H^{min}, V(H) \setminus S_H^{min})$ be the balanced min-cuts of $V(H)$. Then, we have that*

$$\sum_{\substack{H \to (S_1, S_2) \\ \text{induced by nodes in } \mathcal{T}}} 2 \cdot w(S_H^{min}, V(H) \setminus S_H^{min}) \cdot |H| \leq O(1) \cdot \mathsf{OPT}_G.$$

To prove the lemma, we need to use a white-box analysis of the charging argument for sparsest cuts and balanced min-cuts as in Charikar and Chatziafratis Charikar & Chatziafratis (2017) (see also, e.g. Agarwal et al. Agarwal et al. (2022) and Assadi et al. Assadi et al. (2022)). In particular, consider any HC tree $\mathcal{T}$ and any edge $(u, v) \in E$, they use the following notion of 'cost footprint' to lower bound the optimal cost.

**Definition 12** (Edge cost footprint). Let $G = (V, E, w)$ be a weighted undirected graph, and let $\mathcal{T}$ be an HC tree of $G$. For any edge $(u, v) \in E$ and integer $t \geq 1$, we say the *cost footprint* of edge $(u, v) \in E$ in $\mathcal{T}$ of size $t$, denoted as $\mathsf{edge\text{-}cost}_{\mathcal{T}}^t((u, v))$, is as follows:

$$\mathsf{edge\text{-}cost}_{\mathcal{T}}^t((u, v)) = \begin{cases} w(u, v), & \text{if } (u, v) \text{ crosses any pair of } \textit{maximal} \text{ clusters of size at most } t \text{ in } \mathcal{T}; \\ 0, & \text{otherwise.} \end{cases}$$

Intuitively, the notion of $\mathsf{edge\text{-}cost}_{\mathcal{T}}^t((u, v))$ captures the "levels" where edge $(u, v)$ would pay non-zero costs. To elaborate, consider an edge $(u, v) \in E$. Before the edge is split, it pays no cost. After the edge is split, and suppose the size of the cluster on which $(u, v)$ is split is $s$, then $(u, v)$ pays exactly a cost of $s \cdot w(u, v)$ to the cost. In Definition 12, note that for any $t \geq s$, we will have $\mathsf{edge\text{-}cost}_{\mathcal{T}}^t((u, v)) = 0$. On the other hand, for any $t \leq s - 1$, since the two clusters that are induced by the edge split of $(u, v)$ are of size at most $(s - 1)$, we have $\mathsf{edge\text{-}cost}_{\mathcal{T}}^t((u, v)) = w(u, v)$ for $t \in [0, s - 1]$. Therefore, the summation $\sum_{t=0}^n \mathsf{edge\text{-}cost}_{\mathcal{T}}^t((u, v))$ exactly characterizes the cost contribution for $(u, v)$ to the HC objective. More formally, this structural property is characterized as follows.

**Lemma G.1** (Charikar and Chatziafratis Charikar & Chatziafratis (2017), cf. Assadi et al. (2022)). *For any graph $G = (V, E, w)$ and any of its HC tree $\mathcal{T}$, there is*

$$\mathsf{cost}_G(\mathcal{T}) = \sum_{t=0}^n \sum_{(u,v) \in E} \mathsf{edge\text{-}cost}_{\mathcal{T}}^t((u, v)).$$

At first glance, working with the notion of edge cost footprint appears to make things more complicated. The advantage of using such a notion is that it allows us to 'charge' the cost of the tree into 'balanced' partitions. In particular, the following technical statements shown by Charikar and Chatziafratis Charikar & Chatziafratis (2017) and Assadi et al. Assadi et al. (2022).

**Claim G.2** (Charikar and Chatziafratis Charikar & Chatziafratis (2017), cf. Assadi et al. (2022)). *Consider the notion of edge cost footprint as defined in Definition 12, we have*

$$\sum_{t=0}^n \sum_{(u,v) \in E} \mathsf{edge\text{-}cost}_{\mathcal{T}}^{2t/3}((u, v)) \leq 3 \cdot \sum_{t=0}^n \sum_{(u,v) \in E} \mathsf{edge\text{-}cost}_{\mathcal{T}}^t((u, v)).$$

The reason for us to work with the value $2t/3$ is that, as we will see later, we can charge the cost of $1/3$-balanced min-cuts to the summation, which gives us a way to control the cost obtained by the HC tree obtained by recursive *approximate* $1/3$-balanced min cut.

Another technical advantage of working with the edge cost footprint is that we can "decompose" the cost into unit terms, and the terms of summation does *not* have to be "synchronized" with the terms we are summing over. The technical claim is as follows.

**Claim G.3** (Charikar and Chatziafratis Charikar & Chatziafratis (2017), cf. Assadi et al. (2022)). *Let* $\mathcal{T}$ *be any HC tree of G. Furthermore, consider any function $F(u, v, t)$ in the summation*

$$\sum_{\substack{H \to (S_1, S_2) \\ \text{induced by nodes in } \mathcal{T} \\ \text{on level } \ell}} \sum_{t=|S_2|+1}^{|H|} \sum_{(u,v) \in E(H)} F(u, v, t).$$

*Then, each $u'$, $v'$ and $t'$ (and the corresponding term $F(u', v', t')$) appear at most once in the summation.*

Note that a challenge for us to bound the term

$$\sum_{\substack{H \to (S_1, S_2) \\ \text{induced by nodes in } \mathcal{T}}} 2 \cdot w(S^{min}, V(H) \setminus S^{min}) \cdot |H|$$

in Lemma F.3 is that the weights $w(S^{min}, V(H) \setminus S^{min})$ and the term we are summing over $(H \to (S_1, S_2))$ are very different from each other. With Claim G.2 and Claim G.3, we can proceed by $i)$. first bound each term of $w(S^{min}, V(H) \setminus S^{min}) \cdot |H|$ by a function of $\text{edge-cost}_{\mathcal{T}}^t((u, v))$ for some $t$ and $(u, v)$ in the optimal HC tree $\mathcal{T}^*$; and $ii)$. handle the summation with Claim G.3 to upper-bound the term.

We now show the main lemma that allows us to 'charge' the cost of balanced cuts to the edge cost footprints of the optimal tree. Although this lemma is inherently implied in Charikar & Chatziafratis (2017); Assadi et al. (2022), we include the proof here for completeness.

**Lemma G.4.** *Let $G = (V, E, w)$ be a weighted undirected graph, and let $\mathcal{T}^*$ be the optimal HC tree of G. Furthermore, let $H \subseteq G$ be any vertex-induced subgraph of G, and*

- *let $H \to (S^{min}, V(H) \setminus S^{min})$ be the split on H induced by a $\frac{1}{3}$-balanced min-cut.*

- *let $H \to (S_1, S_2)$ be the split on H induced by any $\frac{1}{3}$-balanced cut.*

*Then, we have that*

$$|H| \cdot w(S^{min}, V(H) \setminus S^{min}) \le 3 \cdot |S_1| \cdot \sum_{(u,v) \in E(H)} \text{edge-cost}_{\mathcal{T}^*}^{2|H|/3}((u, v))$$

$$\le \sum_{t=|S_2|+1}^{|H|} \sum_{(u,v) \in E(H)} \text{edge-cost}_{\mathcal{T}^*}^{2|H|/3}((u, v)).$$

*Proof.* The proof follows essentially from the same argument as in two previous works Charikar & Chatziafratis (2017); Assadi et al. (2022). Let $S_1^*, S_2^*, \cdots, S_k^*$ be the maximal clusters of size at most $\frac{2}{3} \cdot |S|$ and with non-empty intersection with $H$. Observe that we can form new clusters $A = \cup_{j \in I_A} S_j^* \cap H$ and $B = \cup_{j \in I_B} S_j^* \cap H$, such that

- $A \cup B$ contains entire vertices in clusters $\cup_j S_j^*$. In other words, $I_A \cup I_B = [k]$; and

- $\max\{|A|, |B|\} \le \frac{2}{3} \cdot |S|$.

Therefore, we can write that

$$|H| \cdot w(S^{min}, V(H) \setminus S^{min}) \le |H| \cdot w(A, B) \quad \text{(by the definition of 1/3-balanced min-cut)}$$

$$\le |H| \cdot \sum_{(u,v) \in E(H)} \text{edge-cost}_{\mathcal{T}^*}^{2|H|/3}((u, v))$$

$$\text{(by the fact that every split edges has to be at level at most } \frac{2}{3} \cdot |H|)$$

$$\le 3 \cdot |S_1| \cdot \sum_{(u,v) \in E(H)} \text{edge-cost}_{\mathcal{T}^*}^{2|H|/3}((u, v)).$$

$$\text{(by the fact that } H \to (S_1, S_2) \text{ is a balanced cut)}$$

Finally, for the second inequality in the lemma statement, note that for any $(u, v)$ and any $t' < t$, there is $\mathsf{edge\text{-}cost}^{t'}_{\mathcal{T}^*}((u, v)) \geq \mathsf{edge\text{-}cost}^{t}_{\mathcal{T}^*}((u, v))$, which gives us the desired statement. $\quad\square$

***Finalizing the proof of Lemma F.3.*** Lemma F.3 now follows from the combination of Lemmas G.1 and G.4 and Claim G.2 and G.3. To elaborate, we now now bound the cost as follows.

$$\sum_{\substack{H \to (S_1, S_2) \\ \text{induced by nodes in } \mathcal{T}}} 2 \cdot w(S^{min}, V(H) \setminus S^{min}) \cdot |H|$$

$$\leq 6 \cdot \sum_{\substack{H \to (S_1, S_2) \\ \text{induced by nodes in } \mathcal{T}}} \sum_{t=|S_2|+1}^{|H|} \sum_{(u,v) \in E(H)} \mathsf{edge\text{-}cost}^{2|H|/3}_{\mathcal{T}^*}((u, v)) \qquad \text{(by Lemma G.4)}$$

$$= 6 \cdot \sum_{t=0}^{n} \sum_{(u,v) \in E} \mathsf{edge\text{-}cost}^{2t/3}_{\mathcal{T}^*}((u, v)) \qquad \text{(by Claim G.3 since the summations are disjoint)}$$

$$\leq 18 \cdot \sum_{t=0}^{n} \sum_{(u,v) \in E} \mathsf{edge\text{-}cost}^{t}_{\mathcal{T}^*}((u, v)) \qquad \text{(by Claim G.2)}$$

$$\leq 18 \cdot \mathsf{cost}_G(\mathcal{T}^*) = 18 \cdot \mathsf{OPT}_G, \qquad \text{(by Lemma G.1)}$$

as desired by the lemma statement. $\hfill$ Lemma F.3 $\square$

# H ADDITIONAL DETAILS ON EXPERIMENTS

All of our experiments are implemented on a Macbook Pro with M2 CPU. In this section, we provide missing details on the dataset, pre-processing parameters and more experimental results.

## H.1 MORE DETAILS ON DATASETS

We first provide the basic statistics of the datasets used in the paper. The number of edges of SBM and HSBM are the average of all generated graphs.

Table 2: Datasets used for experiments

| Dataset | Synthetic | | Real-world | | |
| --- | --- | --- | --- | --- | --- |
| | SBM | Hierarchical SBM | IRIS | WINE | BOSTON |
| Size | 150 | 150 | 150 | 178 | 506 |
| # Edges | 2614 | 4005 | 4851 | 11830 | 95566 |
| # Clusters | 5 | 5 | 3 | 3 | 5 |

The similarity graph of each real-world dataset is constructed by Gaussian kernel, according to the standard heuristic. The parameter $\sigma$ is selected as 0.65 for WINE and BOSTON, and 5 for IRIS.

## H.2 MORE RESULTS ON SYNTHETIC DATA

The synthetic graphs we use in Section 4 have five clusters ($k = 5$) with sizes of $\{20, 20, 30, 30, 50\}$ for each cluster. Here we demonstrate more experiment results on SBM graphs with other values of $k$ and sizes of each cluster. We give details together with the illustrations.

We start with $k = 6$ SBM graphs, but in two different scenarios: the graphs can have clusters of the same size of various sizes. We set the privacy parameters same as before. The intra-probability is 0.7 ($p = 0.7$) and inter-probability is 0.1 ($q = 0.1$). Similarly as before, our algorithm outperforms input perturbation significantly on all choices of $\varepsilon$. Further, the cost given by our algorithm approaches the cost in the non-private setting. The details are shown in Figure 5 and Figure 6.

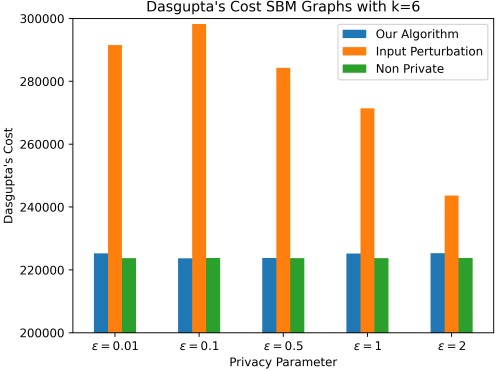

Figure 5: Comparison of Dasgupta's cost on SBM graphs of size $n = 180$ and $k = 6$. Each cluster has the same size of 30.

Figure 6: Comparison of Dasgupta's cost on HSBM graphs of size $n = 180$ and $k = 6$. Clusters have different sizes: 10,20,30,30,40,50.

With the chosen parameters the SBM graphs are well-clustered. In the next set of results we weaken the cluster pattern by reducing $p$ to half. Other settings follow as above.

As shown in Figure 7 and Figure 8, we move on to $k = 8$ with weaker cluster patterns. Observation follows immediately that our algorithm still produces favorable results consistently. Up to this point, we have tested synthetic graphs on a diversity of settings where the number of clusters, cluster sizes and clustering pattern vary, these additional experiment results show that our algorithm can generate private hierarchical clustering with much lower cost comparing to the input perturbation. Further, if

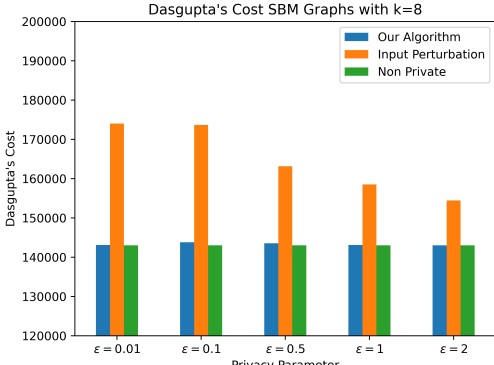 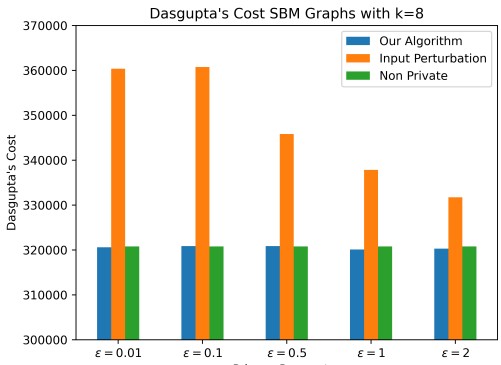

Figure 7: Comparison of Dasgupta's cost on SBM graphs of size $n = 160$ and $k = 8$. Each cluster has the same size of 20.

Figure 8: Comparison of Dasgupta's cost on HSBM graphs of size $n = 200$ and $k = 8$. Clusters have different sizes: 10,10,20,20,30,30,40,40.

the input graph has a well-clustered structure, our algorithm is even comparable to the non-private algorithm in terms of Dasgupta's cost.

