# OpenReview forum: "On the Price of Differential Privacy for Hierarchical Clustering"
_ICLR.cc/2025/Conference — ICLR 2025 Poster_

### Official Review · Reviewer_8Hdn · 2024-10-31

**Soundness:** 3
**Presentation:** 3
**Contribution:** 2
**Rating:** 8
**Confidence:** 3

**Summary:**

This paper addresses the challenge of performing hierarchical clustering (HC) on a graph while ensuring differential privacy (DP). Traditional approaches to differentially private clustering on graphs, particularly under edge-level DP, have recently been shown to incur high errors, which limits their practicality. To improve upon this, the paper proposes an alternative approach using weight-level DP, where only the edge weights (not the topology) represent sensitive information.

While weight-level DP is less commonly studied than edge-level DP, it is not entirely novel. Previous works have defined neighboring graphs under weight-level DP by assuming identical topologies but allowing edge weight differences up to an $\ell_1$ norm of 1. In this study, the standard weight-DP definition is adapted by adding a unit-weight assumption, requiring all edges to have weights of at least 1. For clarity, this adapted model will be referred to as “weight-DP-bis”. This adjustment enables significantly more practical error bounds, which are leveraged in the main contributions.

First, the paper introduces a polynomial-time algorithm that achieves an approximation of $O(\log^{1.5} n / \epsilon) $ for hierarchical clustering under the $\epsilon$-weight-DP-bis model. The paper further justifies the unit-weight assumption in weight-DP-bis by demonstrating that without it, an HC algorithm that satisfies $\epsilon$-weight-level DP must incur an additive error of at least $\Omega(n^2 / \epsilon)$.

As an additional contribution, new lower bounds are established for balanced sparsest cuts under weight-level DP, which may have implications for broader applications beyond HC.

Finally, empirical results support the algorithm’s effectiveness on both synthetic and real datasets, showing performance close to non-private benchmarks and scalability across a range of graph sizes.

**Strengths:**

The paper addresses a significant and complex topic that holds potential benefits for the community. The work is well written, well organized and, based on my level of reading, presents a sound approach to the problem. The core concept is clearly presented, and the technical contributions do not look trivial to me. Overall, the presentation and flow of ideas make the paper engaging and accessible (with a small criticism that I provide below), providing a very nice reading experience.

**Weaknesses:**

I have a few points that could be regarded as potential areas for improvement or clarification in the paper. Please note that these comments are based on my understanding of the material, and they reflect my personal perspective. I do not claim to be a specialist specifically in hierarchical clustering (HC). It is possible that I have misunderstood some aspects, and I am certainly open to revisiting my viewpoint during the discussion phase if necessary.

- **(Minor)**: First, while the paper is quite well written, it may benefit from being more accessible to a general audience. Specifically, it would be helpful to provide brief reminders of key graph-theoretical concepts earlier in the paper. Some concepts, such as sparsest cuts, are not introduced in the main body, while others, such as HC and Dasgupta’s cost, are defined only on page 5. Given that the introduction spans four pages, it may help readers unfamiliar with these concepts to have them presented earlier or concisely defined within the introduction itself to avoid missing key discussions in the first half of the paper. Additionally, the introduction could be shortened to accommodate these clarifications.

- **(Important)**: Second, while I understand that the unit-weight assumption offers better error bounds from a technical standpoint, it may fundamentally modify the problem. Specifically, assuming the $\ell_1$ norm of differences is less than 1 and simultaneously that each edge has a minimum weight of 1, means that two graphs are neighboring only if their weight functions differ on a single edge. As I see it, this implicitly combines aspects of both the $\ell_1$ and $\ell_0$ norms in the privacy definition. The paper may benefit from contextualizing this adapted definition with practical scenarios beyond the initial one presented in [1] (and paraphrased on line 77), as these may no longer directly apply. Additionally, there are existing variations in weight-level DP definitions that might be interesting to discuss in the paper, where the $\ell_1$ norm is that replaced with the $\ell_\infty$ norm. Relevant references include [2,3,4,5].

- **(Important)**: Third, it seems that the paper does not compare to a closely related problem: computing minimum spanning trees under weight-level DP. This problem has been studied under both $\ell_1$ and $\ell_\infty$ norms in [1,2,3]. I believe that this line of work is relevant to this paper because, as far as I understand, computing a weighted minimum spanning tree can be closely related to the single linkage HC algorithm. Existing studies on this problem provide upper and lower bounds that may be highly relevant here, even if the metric used (e.g., not explicitly Dasgupta's cost) differs. For instance, [1,3] provide matching upper and lower bounds of $\Theta(n^2 / \epsilon)$ for the additive error in minimum spanning tree computation under weight-level DP with $\ell_1$ norm, which seems related to Theorem 3 in this paper. It think it would be nice if the paper could discuss this connection and, if relevant, compare these results. Similarly [2] adapts the initial definition to use an $\ell_\infty$ norm and shows that, similar results hold in the $\ell_\infty$ model, but can be circumvented by forcing the $\ell_\infty$ norm to be smaller than $\frac{1}{\vert E \vert}$ in the privacy definition. This scheme seems similar to the idea of modifying the neighborhood definition to obtain better errors, but as pointed out in [3], this can be considered as being an alchemical fix.

- **(Medium)**: Finally, the experimental section could be strengthened by including more comparisons to existing methods or baselines. Currently, it appears that the only baseline comparison is with iterative sparsest cut after input perturbation. It might be beneficial to consider additional comparisons, such as those in Imola (2023), which compares against single and average linkage.

- **(Minor)**: A brief stylistic note: while this is a personal preference, I would suggest avoiding exclamation points in scientific writing, as seen in lines 160 and 200. A more neutral tone may be more fitting for a formal context.



[1] Shortest Paths and Distances with Differential Privacy: https://arxiv.org/abs/1511.04631

[2] Minimum spanning tree release under differential privacy constraints: https://arxiv.org/abs/1801.06423

[3] Near-Universally-Optimal Differentially Private Minimum Spanning Trees: https://arxiv.org/abs/2404.15035

[4] Faster Private Minimum Spanning Trees: https://arxiv.org/abs/2408.06997

[5] Graph-based clustering under differential privacy: https://arxiv.org/abs/1803.03831

**Questions:**

Please comment an the above weaknesses (especially important ones).

---

> ### Author Response · Authors · 2024-11-14
> **Thanks for the review & inquires about the papers you pointed out**
>
> Thanks for the positive feedback and the detailed review. We noticed that you mentioned several papers ([1-5]) in your review, but it appears that the titles of them are not posted. As such, could you provide the titles of these papers? This will help us address your concerns and revise the paper. Thanks!

---

> > ### Comment · Reviewer_8Hdn · 2024-11-15
> >
> > Dear authors,
> >
> > Sorry for my forgetting to upload the references, my review just got updated.
> >
> > Best regards

---

> > > ### Author Response · Authors · 2024-11-21
> > > **Part I of the response**
> > >
> > > > First, while the paper is quite well written, it may benefit from being more accessible to a general audience. Specifically, it would be helpful to provide brief reminders of key graph-theoretical concepts earlier in the paper. Some concepts, such as sparsest cuts, are not introduced in the main body, while others, such as HC and Dasgupta’s cost, are defined only on page 5. Given that the introduction spans four pages, it may help readers unfamiliar with these concepts to have them presented earlier or concisely defined within the introduction itself to avoid missing key discussions in the first half of the paper. Additionally, the introduction could be shortened to accommodate these clarifications.
> > >
> > > We thank the reviewer for the positive feedback on the quality of writing. We also agree with the comment that adding informal definitions for sparsest cuts, HC, and Dasgupta’s cost in an earlier place would help readers with a broader background. We have made changes in the updated version of the manuscripted.
> > >
> > >
> > > > Second, while I understand that the unit-weight assumption offers better error bounds from a technical standpoint, it may fundamentally modify the problem. Specifically, assuming the $\ell_1$ norm of differences is less than 1 and simultaneously that each edge has a minimum weight of 1, means that two graphs are neighboring only if their weight functions differ on a single edge. As I see it, this implicitly combines aspects of both the $\ell_1$ and $\ell_0$ norms in the privacy definition. The paper may benefit from contextualizing this adapted definition with practical scenarios beyond the initial one presented in [1] (and paraphrased on line 77), as these may no longer directly apply.
> > >
> > > We agree that adding more context to the motivation is helpful for our algorithm. We have discussed applications like transaction networks in our paper, and in the new version, we have expanded the discussion by adding the example of the iTunes library. It is a bipartite graph where music/artist are on one side and users on the other side. We know the graph topology because the album brought by the user is known to Apple. However, the number of times a user has interacted with a particular song is private information. Here, the interaction counter starts with 1 because a user has downloaded the album. Hierarchical clustering is a natural problem for this setting.
> > >
> > > The most straightforward way to define neighboring weights is to say exactly one edge changes by $1$ in our model. This case corresponds to neighboring graphs with both $\ell_0$ and $\ell_1$ norms, separately. However, our model is more general: it actually allows change of weights on more than a single edge, and the only requirement for the weight-neighboring model is that the $\ell_1$ norm could change by at most $1$. In fact, as in our response to reviewer ibaF, if the minimum weight of the graph is $\alpha$, our algorithm could get $O(\frac{\log^{1.5}{n}}{\varepsilon \alpha})$-multiplicative approximation. We focused on the case of min-weight as $1$ since we believe the case is most natural for many applications.

---

> > > > ### Author Response · Authors · 2024-11-21
> > > > **Part II of the response due to character limits**
> > > >
> > > > > Additionally, there are existing variations in weight-level DP definitions that might be interesting to discuss in the paper, where the $\ell_1$ norm is that replaced with the $\ell_\infty$ norm. Relevant references include [2,3,4,5].
> > > >
> > > > > Third, it seems that the paper does not compare to a closely related problem: computing minimum spanning trees under weight-level DP. This problem has been studied under both $\ell_1$ and $\ell_\infty$ norms in [1,2,3]. I believe that this line of work is relevant to this paper because, as far as I understand, computing a weighted minimum spanning tree can be closely related to the single linkage HC algorithm. Existing studies on this problem provide upper and lower bounds that may be highly relevant here, even if the metric used (e.g., not explicitly Dasgupta's cost) differs. For instance, [1,3] provide matching upper and lower bounds of $\Theta(n^2/epsilon)$ for the additive error in minimum spanning tree computation under weight-level DP with norm, which seems related to Theorem 3 in this paper. It think it would be nice if the paper could discuss this connection and, if relevant, compare these results. Similarly [2] adapts the initial definition to use an norm and shows that, similar results hold in the $\ell_\infty$ model, but can be circumvented by forcing the $\ell_\infty$ norm to be smaller than $\frac{1}{|E|}$ in the privacy definition. This scheme seems similar to the idea of modifying the neighborhood definition to obtain better errors, but as pointed out in [3], this can be considered as being an alchemical fix.
> > > >
> > > > (We provide a combined response for the MST and the $\ell_{\infty}$ norm)
> > > > We agree that MST is closely related to the hierarchical clustering problem, and we thank you for the suggested references! While the references are closely related, we do not believe that their results and our results could directly imply each other. The error induced by privacy mechanisms for the MST problem is easier to handle since the noises across edges do not accumulate. As such, for the MST problem, it is easy to get $\tilde{\Theta}(n/\varepsilon)$ for $\ell_1$ norm and $\tilde{\Theta}(n^2/\varepsilon)$ for $\ell_\infty$ norm. We have included this discussion in the updated version.
> > > >
> > > > An additional response to the $\ell_\infty$ norm neighboring graphs: for cut-related application (e.g., hierarchical clustering), the neighboring graphs with $\ell_\infty$ norm at most $1$ could change the entire set of ‘important edges’ of the graph. For instance, we could consider two graphs $G_1, G_2$ with different planted cliques, and the weights of the clique edges are 1 while the weight of all other edges are $\delta\rightarrow 0$. Essentially, any low-cost tree on $G_1$ must have $\Omega(n^3)$ additive error compared to the optimal HC tree for $G_2$. In this way, it appears that we could obtain an additive error lower bound of $\Omega(n^3)$, which is very pessimistic.
> > > >
> > > > > Finally, the experimental section could be strengthened by including more comparisons to existing methods or baselines. Currently, it appears that the only baseline comparison is with iterative sparsest cut after input perturbation. It might be beneficial to consider additional comparisons, such as those in Imola (2023), which compares against single and average linkage.
> > > >
> > > >
> > > > Thanks for the valuable suggestion. We have included the linkage-based algorithms in our new version of the paper. The experiments showed that the linkage-based algorithms are worse than ours. This is also consistent with the empirical results of Imola et al. [ICML’23].
> > > >
> > > > > A brief stylistic note: while this is a personal preference, I would suggest avoiding exclamation points in scientific writing, as seen in lines 160 and 200. A more neutral tone may be more fitting for a formal context.
> > > >
> > > >
> > > > Thanks for the comment. we have made changes accordingly.

---

> > > > > ### Comment · Reviewer_8Hdn · 2024-11-21
> > > > >
> > > > > Thank you for your feedback.
> > > > >
> > > > > I believe your response addresses most of my concerns. While I am still somewhat puzzled by edges having a minimum weight of 1, I recognize that this assumption is explicitly stated and not hidden. Therefore, I trust that readers can form their own opinions on this matter.
> > > > >
> > > > > With that in mind, I am pleased to increase my rating.

---

> > > > > > ### Author Response · Authors · 2024-11-21
> > > > > >
> > > > > > Thank you for your thoughtful response and updates. We appreciate your feedback and support!

---

### Official Review · Reviewer_qT5t · 2024-11-04

**Soundness:** 2
**Presentation:** 4
**Contribution:** 4
**Rating:** 6
**Confidence:** 3

**Summary:**

This paper studies the challenge of achieving differential privacy in hierarchical clustering, under a model that assumes edge weights as sensitive information. The authors successfully show there exists an algorithm in this setting that achieves $O(\log^{1.5} n/\epsilon)$
multiplicative approximation.

**Strengths:**

1. Despite the complicated problem setting and technical components, this paper has a clear presentation.
2. In addition to establishing an upper bound, the authors also derive a matching lower bound that aligns with previous work, which presents an additive error of $\Omega(n^2/ \epsilon)$ and  a new lower bound of $\Omega(1/ \log ^{2}n \epsilon)$  for balanced sparsest cuts in the weight-level DP model. These findings offer valuable insights into the comparison between edge-level and weight-level differential privacy models.  The lower bound proofing technique is novel.

**Weaknesses:**

Currently, I do not see the reason why in the input graph G adding an extra additive weight of $ 10 \log \frac{n}{\epsilon} $ would be necessary, can the authors give some comments on this?

**Questions:**

Please see above.

---

> ### Author Response · Authors · 2024-11-21
> **The additive weights are necessary for correctness**
>
> > Currently, I do not see the reason why in the input graph G adding an extra additive weight of $10\log\frac{n}{\varepsilon}$ would be necessary, can the authors give some comments on this?
>
> Thanks for the question. We provided the intuitions of why the additive weights are necessary in section 1.1, and we provide a more concise version of the explanation here. The cost of the HC tree is basically proportional to the error on the sparsest cut. If we do not perform the additive weights and instead directly add Laplace noise, then some other cuts (that are far from the sparest) may appear to be the sparsest due to the noises. As such, we cannot effectively control the additive error
>
> Another way to see this is through our experiments. Here, if the additive weights are not introduced, the algorithm is essentially input perturbations. It could be observed that our algorithm performs way better than input perturbations on both synthetic and real-world datasets.

---

> > ### Author Response · Authors · 2024-11-25
> >
> > Hi Reviewer qT5t, we wanted to check whether you had any other concerns, thanks!

---

### Official Review · Reviewer_ibaF · 2024-11-04

**Soundness:** 3
**Presentation:** 3
**Contribution:** 3
**Rating:** 6
**Confidence:** 4

**Summary:**

The paper proposes several results furthering hierarchical clustering with differential privacy. The notion of privacy is weighted differential privacy, where the graph topology is public, and the weights of the edges are private. First, an efficient algorithm obtaining a \frac{n}{\epsilon} \log n approximation ratio for graphs in which the minimum edge weight is at least 1 is proposed. The authors show that in the worst case, the \omega(n^2 / \epsilon) additive error lower bound from previous work still holds for the weaker notion of weighted DP. They make the interesting observation that the lower bound carries over to DP balanced cut via a reduction. Finally, they run experiments showing that the proposed mechanism is indeed feasible.

**Strengths:**

The proposed algorithm is simpler and much more efficient than that in prior work. The multiplicative error is not too high for the graphs with edge weight at least 1, which is also a nice result and improvement over prior work. The reduction to balanced cut is an interesting new result.

**Weaknesses:**

The lower bound in a way misses the spirit of the proposed algorithm: It shows that there are adversarial graphs (with edge weights possibly zero) which require additive \Omega(n^2 / \epsilon) noise, but the upper bound applies to graphs whose edge weights are all at least 1. As these types of graphs are central to obtaining the upper bound, it would be more interesting to see what the lower bound is for graphs which also satisfy this property.

The utility of the proposed algorithm for graphs whose edge weights are close to zero is currently undefined, meaning the algorithm is not robust for graphs which fall even a little outside the constraints. This can be reconciled somewhat by combining this algorithm with previous work, but then the efficiency improvements do not hold.

The experiments look interesting, though there is an unexplained phenomenon that the proposed algorithm barely shows any sensitivity to epsilon: it achieves nearly the same error on both the minimum \epsilon = 0.1 and maximum \epsilon = 2.0 tested. This is surprising, since typically epsilon dramatically affects the performance of any private algorithm, and typically as \epsilon becomes large, the cost should approach the cost of the best tree. Neither of these things is happening in the current plots; I think they should be double-checked.

**Questions:**

Can you develop a lower bound family of graphs on a fixed topology which all have edge weights above 1, and incur at least the stated multiplicative error?

Is it possible to obtain a stand-alone utility theorem for the proposed algorithm, which interpolates between \log n / \epsilon multiplicative error and n^2 / \epsilon additive error for graphs with edge weights at least 1 vs. arbitrary graphs? You might have to "unpackage" Proposition A.6 so that it can handle additive error.

For the experiments, what is the smallest value of epsilon for which the utility of the proposed algorithm becomes "bad"? What is the value of epsilon such that the utility approaches the optimal cost? Consider increasing the range of epsilons used in the plots.

---

> ### Author Response · Authors · 2024-11-21
>
> > The lower bound in a way misses the spirit of the proposed algorithm: It shows that there are adversarial graphs (with edge weights possibly zero) that require additive \Omega(n^2 / \varepsilonilon) noise, but the upper bound applies to graphs whose edge weights are all at least 1. As these types of graphs are central to obtaining the upper bound, it would be more interesting to see what the lower bound is for graphs that also satisfy this property.
>
> We want to clarify that the purpose of our lower bound is to provide a technical justification for our privacy model. The lower bound says that without the additional unit weight condition of the privacy model, it is hard to get a satisfactory approximation even with weight-neighboring graphs.
>
> We agree that an HC lower bound with our privacy model is interesting. However, such lower bounds appear to be hard to obtain with the 5-cycle instance family. Polylog multiplicative approximation is usually considered ‘the sweet spot’ in the literature of HC, which is part of the reason we did not aim for further improvements. Currently, we do not rule out potential $O(1)$-approximation with our model. We agree that the suggested lower bound (or a potential algorithm) is an interesting direction to pursue in the future.
>
>
> > The utility of the proposed algorithm for graphs whose edge weights are close to zero is currently undefined, meaning the algorithm is not robust for graphs which fall even a little outside the constraints. This can be reconciled somewhat by combining this algorithm with previous work, but then the efficiency improvements do not hold.
>
> Thanks for pointing this out. By our analysis, it follows that if the minimum weight is at least $\alpha$, our algorithm could get $\varepsilon$-DP HC with $O(\frac{\log^{1.5}{n}}{\varepsilon \alpha})$-multiplicative approximation. That is, our algorithm could indeed handle the case you mentioned, and it is quite robust since the approximation degrades inversely with $\alpha$. This is an excellent point, and we have added a section about this in the updated version of the manuscript.
>
> > The experiments look interesting, though there is an unexplained phenomenon that the proposed algorithm barely shows any sensitivity to epsilon: it achieves nearly the same error on both the minimum \varepsilonilon = 0.1 and maximum \varepsilonilon = 2.0 tested. This is surprising, since typically epsilon dramatically affects the performance of any private algorithm, and typically as \varepsilonilon becomes large, the cost should approach the cost of the best tree. Neither of these things is happening in the current plots; I think they should be double-checked.
>
> This is indeed an interesting point, and we should have discussed it in our paper. We are confident that there is no error or bug in the implementation. The reason for the surprising phenomenon is that unlike conventional DP algorithms that release **values**, our algorithm releases the *HC trees*. The final cost is obtained by cost evaluation on the original graph. For the privately computed HC tree, even if we use a smaller $\varepsilon$ value, the *structure* of the weights in the graph is mostly preserved by our algorithm.
>
>
> > Can you develop a lower bound family of graphs on a fixed topology which all have edge weights above 1, and incur at least the stated multiplicative error?
>
> As we have mentioned in our response to the ‘weakness’ section, such a lower bound appears hard to obtain, and currently, it might be possible to get $O(1)$-multiplicative approximation. On the other hand, $\text{polylog}(n)$-multiplicative approximation to Dasgupta’s cost is usually regarded as ‘sufficiently good’ in the HC literature.
>
>
> > Is it possible to obtain a stand-alone utility theorem for the proposed algorithm, which interpolates between \log n / \varepsilonilon multiplicative error and n^2 / \varepsilonilon additive error for graphs with edge weights at least 1 vs. arbitrary graphs? You might have to "unpackage" Proposition A.6 so that it can handle additive error.
>
> We are not sure what you mean about the interpolation between the $\log{n}/\varepsilon$ multiplicative and the $n^2/\varepsilon$ additive error. In the worst case, the additive error of our algorithm has to be $\Omega(n^3)$: the error is attained by a complete graph with unit weights.
>
> If the review means an algorithm that ‘adjusts’ the cost guarantee based on the instances (i.e., in some case $\log n / \varepsilon$-multiplicative approximation, and in some case $O(n^2 / \varepsilon$ additive error, whichever is smaller), then we already gave this result in Lemma D.3 and D.4 (in the appendix).
>
> On a side note, we do have an ‘unpackaged’ analysis of Proposition A.6 in our proof for the lower bound of the balanced sparsest cuts. We do not believe this alone could lead to the case-wise error guarantees, though.

---

> > ### Author Response · Authors · 2024-11-21
> > **Continued responses due to character limits**
> >
> > > For the experiments, what is the smallest value of epsilon for which the utility of the proposed algorithm becomes "bad"? What is the value of epsilon such that the utility approaches the optimal cost? Consider increasing the range of epsilons used in the plots.
> >
> > Thanks for the question. We expand the range of epsilon to [0.001, 100] and test our algorithm on the synthetic SBM graphs. The average additive cost by our algorithm compared to the non-private algorithm is reported below.
> >
> > |                         | 0.001-0.01 | 0.01-0.1 | 0.1-1 | 1-10 | 10-100 |
> > |---------------------|----------------|-------------|--------|--------|-----------|
> > |  Additive error  |      310       |      72     |    19  |   5.5  |    2.0    |
> >
> > We could observe that the additive error decreases relatively fast as the value of epsilon grows. And after epsilon is taken as a large constant, the additive cost is very small. Note that we mentioned in the paper that for well-clustered graphs, our algorithm works well and sometimes performs on-par to the non-private algorithm, as reported in the paper (Figure 1, even epsilon >=0.01).
> >
> > Theoretically, the smaller epsilon is, the worse the utility can become. We also tested a few rare scenarios where epsilon is extremely small (0.00001) and observed that the additive error is indeed larger on average (>1000).

---

> ### Comment · Reviewer_ibaF · 2024-11-22
> **Thanks for your responses**
>
> I appreciate the addition of Appendix E, it concretely shows the multiplicative approximation ratio as the edge weight goes to 0.
> My original intent of the comment was to suppose that just a small number of edges in the graph have weight 0, which seems to be another possible violation of the assumption. In this case Proposition E.1 still does not provide a guarantee. Also, I appreciate the additional experimental data, and I do agree that the algorithm appears to be finding the correct tree even at very small values of epsilon.
>
> Overall, I find the assumption that all edge weights >1 to be slightly unnatural. However, I see the theoretical and experimental merits of this work, and thus I will update my score.

---

> ### Author Response · Authors · 2024-11-22
>
> Thanks for getting back to us and for the support!
>
> A short response to your follow-up comment: we agree that Proposition E.1 cannot handle the case when a number of edges actually have $0$ weights. The challenge here is to get a purely multiplicative approximation, we need a lower bound on the weights of the sparsest cuts. Here, even if only a few edges have $0$ values, the weights of the sparsest cut can still go to $0$, which creates an unbounded multiplicative gap. We believe it requires additional assumptions to handle this challenge, and it is an interesting direction to explore for future research.

---

### Author Response · Authors · 2024-11-21

We thank the reviewers for their insightful questions and valuable feedback. We also appreciate the positive comments on the paper, including:

- The proposed algorithm is simpler and much more efficient than that in prior work. (Reviewer ibaF)
- The multiplicative error is not too high for the graphs with edge weights of at least 1, which is also a nice result and improvement over prior work. (Reviewer ibaF)
- The reduction to balanced cut is an interesting new result. (Reviewer ibaF)
- Despite the complicated problem setting and technical components, this paper has a clear presentation. (Reviewer qT5t)
- In addition to establishing an upper bound, the authors also derive a matching lower bound that aligns with previous work, which presents an additive error of $\Omega(n^2/\varepsilon)$ and a new lower bound of $\Omega(1/\log^2 n \, \varepsilon)$ for balanced sparsest cuts in the weight-level DP model. (Reviewer qT5t)
- These findings offer valuable insights into the comparison between edge-level and weight-level differential privacy models. (Reviewer qT5t)
- The lower bound proofing technique is novel. (Reviewer qT5t)
- The paper addresses a significant and complex topic that holds potential benefits for the community. (Reviewer 8Hdn)
- The work is well written, well organized and, based on my level of reading, presents a sound approach to the problem. (Reviewer 8Hdn)
- The core concept is clearly presented, and the technical contributions do not look trivial to me. (Reviewer 8Hdn)
- Overall, the presentation and flow of ideas make the paper engaging and accessible (with a small criticism that I provide below), providing a very nice reading experience. (Reviewer 8Hdn)

We have uploaded an updated version of the manuscript, which incorporates the reviewers' feedback, with changes marked in **red**. Specifically, we have added:
- Additional experiments comparing to more baselines, in particular, linkage-based approaches.
- Informal intuition on central concepts such as HC, Dasgupta’s objective, and sparsest cut, prior to their formal definition in the subsequent text to make the paper more accessible to the general audience.
- Discussion on additional related works, such as minimum spanning trees under weight-level DP.
- An additional discussion about the performance of our algorithm on graphs with minimum weights less than $1$.

We believe these revisions have strengthened the overall paper and we look forward to further feedback. We provide specific detailed responses to individual reviewer comments below.

---

### Meta-Review · Area_Chair_9jWt · 2024-12-24

**Metareview:**

The paper considers Hierarchical clustering (with Dasgupta's objective) in the weight Differential Privacy framework. Here the underlying graph topology is known and the weights need to be protected (in the sense of DP). While the weight-privacy model has been studied before, it is something that could be justified more directly in the paper. The algorithm is simpler compared to prior work and the authors show why the unit-weight assumption is crucial, in that otherwise very strong lower bounds can be constructed. There are also some empirical results. There remain open questions about tightness of the results within the unit-weight DP setting, but these may well be difficult questions and suitable for future work.

**Additional Comments On Reviewer Discussion:**

The reviewers engaged well with authors. There was not much discussion post author rebuttal.

---

### Decision · Program_Chairs · 2025-01-22

Accept (Poster)